# Evolution of multivariate drought hazard, vulnerability and risk in India under climate change

Venkataswamy Sahana[1,3], Arpita Mondal[1,2]

[1]Department of Civil Engineering, Indian Institute of Technology Bombay, Powai, Mumbai 400076, India

[2]Interdisciplinary Program in Climate Studies, Indian Institute of Technology Bombay, Powai, Mumbai 400076, India

*Correspondence to*: Arpita Mondal (marpita@civil.iitb.ac.in)

**Abstract.** Changes in climate and socio-economic conditions pose a major threat to water security, particularly in the densely-populated, agriculture-dependent and rapidly developing country of India. Therefore, for cogent mitigation and adaptation planning, it is important to assess the future evolution of drought hazard, vulnerability and risk. Earlier studies demonstrate

projected drought risk over India on the basis of frequency analysis and/or hazard assessment alone. This study investigates and evaluates the change in projected drought risk under future climatic and socio-economic conditions by combining drought hazard and vulnerability projections at a country-wide scale. A multivariate standardized drought index (MSDI) accounting for concurrent deficits in precipitation and soil moisture is chosen to quantify droughts. Drought vulnerability assessment is carried out combining exposure, adaptive capacity and sensitivity indicators, using a robust multi-criteria decision-making

method called the Technique for Order Preference by Similarity to an Ideal Solution (TOPSIS). In the worst-case scenario for drought hazard (RCP2.6-Far future), there is a projected decrease in the area under high or very high drought hazard classes in the country by approximately 7%. Further, the worst-case scenario for drought vulnerability (RCP6.0-SSP2-Near future) shows a 33% rise in the areal extent of high or very high drought vulnerability classes. West Uttar Pradesh, Haryana and West Rajasthan regions are found to be high risk under all scenarios. Bivariate choropleth analysis shows that the projected drought

risk is majorly driven by change in drought vulnerability attributable to societal developments, rather than changes in drought hazard resulting from climatic conditions. The present study can aid policy makers, administrators and drought managers in developing decision support systems for efficient drought management.

## 1 Introduction

Droughts play a major role in water resources planning and management, agronomy and freshwater availability (Mishra and

Singh, 2010, 2011). Droughts may be exacerbated by climate change or societal developments or by a combination of the two. For building drought resilience, it is important to assess the role of these changes on the evolution of drought at regional scales, particularly for rapidly-growing heavily agriculture-dependent countries such as India. Though socio-economic development is reported to have a greater impact on the water availability as compared to the climate induced impacts in some regions across the globe, the role of climate change cannot be entirely eliminated (Koutroulis et al., 2019a). Representative Concentration

Pathways (RCPs; van Vuuren et al., 2011) that are radiative forcing scenarios for different greenhouse gas emission levels are commonly used for climate change impact studies. Shared Socio-economic Pathways (SSPs; O'Neill et al., 2017), on the other hand, provide different narratives of future societal development. Plausible combinations of different RCPs and SSPs are useful to study the future projections of drought risk (Kim et al., 2020).

    According to the Intergovernmental Panel on Climate Change (IPCC) Fifth Assessment Report (AR5) (IPCC, 2014), risk of

an extreme event can be quantified as a product of hazard, vulnerability and exposure. Drought hazard is a function of magnitude and occurrence probability of drought events. On the other hand, drought vulnerability is the degree to which a region is susceptible to drought and is a function of sensitivity, adaptive capacity and exposure components. These components in turn describe the socio-economic, physical and infrastructural factors and are illustrated through drought vulnerability indicators. A comprehensive drought risk assessment involves proper selection of drought indicators for hazard analysis and

proper selection of drought vulnerability indicators and reliable aggregation technique for vulnerability analysis (Carrão et al., 2016; Naumann et al., 2014; Sahana et al., 2021). By virtue of taking into consideration both drought hazard and vulnerability, a combination of RCP and SSP scenarios offer a comprehensive approach for drought risk projection.

    Several studies have carried out risk assessment of drought and water availability across different regions of the world under changing climate and socio-economic conditions. Singh & Kumar (2019) quantified the water availability in the Indian region

due to climate and demographic changes. Ahmadalipour et al. (2019) carried out drought risk assessment in the African region for different population growth and climate change scenarios. Chen et al. (2021) evaluated the effect of changing climate, population and GDP on the drought risk for China. Park et al. (2020) presented drought risk projections under changing meteorological conditions and socio-economic scenarios for South Korea. A comprehensive drought risk assessment for Europe was carried out by Koutroulis et al. (2018) under changing climate and socio-economic scenarios by evaluating

exposure, sensitivity and adaptive capacity components for the projected period. Along similar lines, Koutroulis et al. (2019) quantified the global water availability under high-end climate change. Water use vulnerability was assessed by Kim et al. (2020) for a river basin in Korea for different climate and socio-economic scenarios.

    For the Indian region, projections of drought hazard/risk or water availability are developed in earlier studies using climate scenarios alone (Aadhar & Mishra, 2020, 2021; Gupta et al., 2020; Gupta & Jain, 2018) with the exception of Singh and Kumar

(2019) who consider the role of both climate and socio-economic scenarios for obtaining future projections of water availability (Singh and Kumar, 2019). However, Singh and Kumar (2019) represent future socio-economic changes using a simplistic approach that considers changes in population alone. A combination of RCP and SSP scenarios by integrating hazard and vulnerability information is required to assess drought risk in India in the Near and Far future. Further, most studies that assess drought hazard under climate change scenarios consider either univariate or multivariate approaches based on precipitation

deficits and temperature effects (Aadhar & Mishra, 2020, 2021; Gupta et al., 2020; Gupta & Jain, 2018). However, droughts can often manifest as a complex interplay of multiple influencing variables necessitating a multivariate approach for

characterization of drought hazard (Sahana et al., 2020). For the agrarian country of India, agro-meteorological drought hazard projections should consider deficits in precipitation or soil moisture or both.

The present study aims at comprehensive drought risk projections for India by accomplishing the following objectives: a) Multivariate drought hazard projection using Multivariate Standardized Drought Index (MSDI) that considers concurrent deficits in precipitation and soil moisture for future warming scenarios. b) Drought vulnerability projection considering combinations of RCP and SSP scenarios, using a list of drought vulnerability indicators that represent exposure, sensitivity and adaptive capacity components. c) Drought risk projection integrating hazard and drought vulnerability information. d) Development of bivariate choropleth plots under future scenarios to quantify the individual roles of climate and societal changes in driving drought risk, and d) Identification of regions and zones that are expected to be under worst drought risk conditions in the Near and Far future.

## 2. Materials and methods

### 2.1 Data

#### 2.1.1 Hydro-climatic variables

The multivariate drought risk assessment focusing on agricultural drought, requires a combined analysis of precipitation as well as soil moisture data. The drought hazard assessment for baseline period (1980-2015) requires observed hydro-climatic variables. Gridded daily precipitation data (mm) at 0.25° lat.× 0.25° lon. resolution is obtained from India Meteorological Department (IMD) (Pai et al., 2014). This dataset has been employed in various studies over Indian region (Sahana et al., 2021). Gridded monthly root-zone soil moisture data ($m^3/m^3$) over the Indian region at 1/2° lat.× 2/3° lon. resolution is obtained from Modern-Era Retrospective Analysis for Research and Application (MERRA-Land). This dataset has been employed for drought studies across the world (Farahmand & AghaKouchak, 2015; AghaKouchak, 2015) and also for Indian regions (Sahana et al., 2020; Sahana et al., 2021). The above two datasets are regridded to a common spatial resolution of 0.5° lat.× 0.5° lon. and rescaled to monthly resolution for the historical drought hazard assessment. Re-gridding of the observed datasets to 0.5° lat.× 0.5° lon resolution is carried out using the Triangulation-based linear interpolation method (Watson and Philip, 1984). Further, monthly time series of spatial variation in terms of standard deviation of precipitation and soil moisture from their observed and rescaled datasets is shown in Figure S1. It is observed that the rescaling of datasets from their parent resolution to 0.5° lat.× 0.5° lon results in no additional variability.

In order to evaluate the projected drought hazard over India, the projected precipitation and soil moisture data at a spatial resolution of 0.5° lat.× 0.5° lon. is obtained from the Inter-Sectoral Impact Model Intercomparison Project (ISIMIP) (Warszawski et al., 2014). The historical (1980-2005) and projected (2006-2099) data from available GCMs namely GFDL-ESM2M, HADGEM2-ES, IPSL-CM5A-LR and MIROC5, and for two RCPs – RCP2.6 and RCP6.0 are downloaded from ISIMIP data portal (https://esg.pik-potsdam.de/search/isimip/). The daily precipitation data (kg m$^{-2}$ s$^{-1}$) is already been

downscaled and bias corrected with respect to global level observed precipitation from EartH2Observe observations, WFDEI and ERA-Interim data Merged and Bias-corrected for ISIMIP (EWEMBI). This data has been previously used to study the soil moisture droughts for Europe (Grillakis, 2019) and terrestrial water storage in mainland China (Jia et al., 2020). The country-wide average annual precipitation for the projected period is higher compared to the baseline periods as shown in Figure S2. As a part of ISIMIP2b experiments, the LPJmL impact model (Sitch et al., 2003), a global vegetation model that is capable of representing fine resolution physical processes using carbon, water and energy balance equations (Schaphoff et al., 2018) under a changed climate, is driven by the bias-corrected GCM precipitation to simulate the root-zone soil moisture (kg m$^{-2}$). For our study, the soil moisture data upto 3 layers accounting for 1 m depth is used. The country-wide average annual soil moisture for the projected period is slightly lower compared to the baseline periods as shown in Figure S3. The observed and simulated country-wide average of monthly precipitation and soil moisture for the period 1980-2005 is presented in Figure S4. The performance of all the ISIMIP models are comparable with that of the observed data, except for the soil moisture during monsoon months. The lowered soil moisture estimates from LPJmL model (ISIMIP experiments) simulations compared to the MERRA-Land soil moisture observations for the monsoon months could be due to overestimation of LPJmL's simulated runoff (Zaherpour et al., 2018). Although the simulated soil moisture data underestimates the monsoon months' soil moisture (June, Jul, Aug, Sep) during the historic period (1980-2005) (Figure S4), we did not perform the bias correction, since we intend to capture the variability in the soil moisture rather than their magnitudes for drought index calculation. The projected daily precipitation is cumulated over each month to get the monthly precipitation values and converted its units from kg m-2 s-1 to mm. The projected monthly soil moisture (average monthly soil moisture) from the model is converted from kg m-2 to m3/m3. The ensemble mean of monthly precipitation and soil moisture from different GCMs is computed. Further, these ensemble mean monthly precipitation and soil moisture time series is used for drought hazard assessment. Although climate variables from CMIP6 are available, drought responses by CMIP5 models are similar to that of CMIP6 (Cook et al., 2020). Hence we proceeded with the CMIP5 data for drought hazard assessment.

## 2.1.2 Drought vulnerability indicators

The country-wide drought vulnerability indicators adopted for drought vulnerability assessment are listed in Table 1, along with their sources, spatial and temporal distribution, units, method of data generation, relevance and correlation to drought vulnerability for both the observed (around the year 2010) and projected datasets (2005-2100). The presented drought vulnerability indicators comprise sensitivity, exposure and adaptive capacity indicators (Table 1). Drought vulnerability indicators such as groundwater availability, irrigation index and waterbody fraction for the projected period are not directly available. Hence, these indicators are proxied by representative indicators (Table 1) through multiple linear regression (MLR). An extensive vulnerability assessment encompasses other social and economic vulnerability indicators such as those used by Meza et al. (2020). However, for a densely-populated and rapidly-developing nation such as India, acquisition of reliable datasets on these indicators is often challenging. Most importantly, unavailability of projections of these indicators over the

Indian region limits their use in this study, since our primary goal is to compare baseline drought risk with that under future projected climate change. Further, the weightages for the categorical vulnerability indicators for drought vulnerability assessment is adopted from (Ekrami et al., 2016; Sahana et al., 2021; Thomas et al., 2016), and is given in Table S1. Finally, drought vulnerability indicators are extracted for the RCP2.6-SSP2 and RCP6.0-SSP2 scenarios for the periods 2060 and 2100 so as to represent different climate and socio-economic scenarios for the Near future and Far future periods respectively. In

general, socio-economic development is a slow process, and takes time to reflect in terms of significant changes in the socio-economic indicators (Dellink et al., 2017). Further, majority of the drought vulnerability/risk studies across the globe have adopted static vulnerability assessment that represent drought vulnerability snapshot in time (Hagenlocher et al., 2019). Therefore, we used the static vulnerability indicators for the year 2010, 2060 and 2099 to quantify drought vulnerability for the baseline, Far future and Near future period respectively.

Drought vulnerability indicators such as population density and GDP for the year 2010 from SSP2 pathway are comparable with their respective observed dataset, with small/negligible difference between the observed and SSP-simulated datasets (Figure S5). Further, drought vulnerability indicators such as groundwater availability, irrigation index and waterbody fraction for the projected period are not directly available. Hence, these indicators are proxied by their representative indicators (Table 1) using multiple linear regression (MLR). Consequently, irrigation ratio, groundwater availability and water body fraction for

the projected period are derived based on relationships between them and the representative variables in the baseline period, and therefore consistency is ensured. The Land Use Harmonization (LUH) (Chini et al., 2014) dataset provides the fractional land use classes for the time period 1500-2100. The historical maps of crop and pasture data from HYDE 3.1 (Hurtt et al., 2011), and estimates of historical national wood harvest and of shifting cultivation are used as input for 1500-2005. Further, the projections of LULC for 2005-2100 are based on the Integrated Assessment Model (IAM) implementations of the RCPs.

Each IAM for different RCPs are used as input to the Earth System Models (ESMs) for future carbon-climate projections. Therefore, LULC scenarios are based on RCPs. LUH is a credible dataset for LULC projection and has been previously used for drought risk projection in South-Asian region (Chou et al., 2019). Further, LULC projections can also be derived based on the land use models, using past LULC data and socio-economic factors driving the land use change. However, development of such models at country scale is beyond the scope of the present study.

**2.2 Methods**

The methodology adopted to study the evolution of drought risk is given in Figure 1.

**2.2.1 Drought hazard assessment**

Drought hazard forms an important component of drought risk assessment. Here, we assess the country-wide drought hazard based on the deficits in precipitation and soil moisture. Therefore, the multivariate standardized drought index (MSDI) of the

non-parametric form is computed using the bivariate case of Gringorten plotting position formula (Gringorten, 1963). MSDI

is equally capable of capturing deficits individually in precipitation or soil moisture, or their joint deficit, considering dependence between these two variables. This is a unique advantage of MSDI (Hao and AghaKouchak, 2014) over other univariate indices. Further, MSDI is capable of representing the onset, propagation and termination of drought. In the Figure S6, considering -0.8 as the threshold for drought trigger, it seen that whenever either the SPI or the SSI falls below this threshold, MSDI covers the critical trajectory and offers a conservative characterization of drought, thereby capturing attenuation and lag effects. The steps involved in the calculation of MSDI is presented below.

1. The joint probability distribution of the 1-month time scale precipitation ($R$) and soil moisture ($S$) is given by

$$P\,(R\,\leq\,r, S\,\leq\,s) = p \tag{1}$$

    where $r$ and $s$ represents the value of the random variables $R$ and $S$ respectively, and $p$ represents the joint probability of the precipitation and soil moisture.

2. For the sample size $n$, the count of occurrence of the pair $(r_i, s_i)$ for $r_i \leq r_k$ and $s_i \leq s_k$ is denoted as $m_k$. $r_k$ and $s_k$ here denote the kth observation for precipitation and soil moisture respectively. The number of joint occurrences ($m_k$) of precipitation and soil moisture pair below $r_k$ and $s_k$ from the whole set of observations is used to calculate empirical joint probability for kth observation based on bivariate Gringorten plotting position (Gringorten, 1963) as

$$P(r_k, s_k) = \frac{m_k - 0.44}{n + 0.12} \tag{2}$$

3. The above empirical joint probability is then standardized to obtain the multivariate index MSDI.

$$MSDI = \varphi^{-1}(P) \tag{3}$$

    where $\varphi$ is the standard normal distribution function. Since the empirical distributions use ranks of data instead of actual values, the sample size should be sufficiently large.

The method of drought hazard assessment followed in the present study is based on Kim et al. (2015). Hazard is measured as the product of magnitude and the associated frequency of occurrence of an event. The MSDI time series at each region is categorized into four groups similar to Mckee et al. (1993). These categories are assigned weights according to the magnitude of MSDI value. Higher weights will be assigned to worst (high negative) MSDI values, and vice versa. Further, each weight category is divided into different clusters based on the frequency of occurrence of MSDI values. The total number of clusters for ratings in each MSDI category is determined using the prominent k-means data clustering algorithm. Higher ratings will be assigned to the cluster with high frequency values, and vice versa. The weightage and rating scheme is depicted graphically in Figure 1. In the k-means clustering technique, distance between the data points is computed using the squared Euclidean distance metric. To avoid the convergence to local minima, the k-means algorithm is run with 100 random initial seeds with 10000 iterations. The Calinski-Harabasz Index (CHI) (Caliński and Harabasz, 1974) is used to determine the optimum number of clusters and is given by

$$CHI = \frac{n-K}{K-1} \times \frac{BGSS}{WGSS} \tag{4}$$

where $n$= number of data points, $K$= number of clusters, $BGSS = \sum_{k=1}^{K} n_k ||G^{\{k\}} - G||^2$ is the between the group scatter, $G^{\{k\}}$ = centroid of the k$^{th}$ cluster, $G$ = centroid of all the observations, $WGSS = \sum_{k=1}^{K} WGSS^{\{k\}}$ is within the group scatter and $WGSS^{\{k\}} = \sum_{i \epsilon I_k} ||M_i^{\{k\}} - G^{\{k\}}||^2$, where $M_i^{\{k\}}$ are the observations. The k-means clustering algorithm is driven for 1 to $n$ clusters. The number of clusters that gives highest value of CHI is the optimum number of clusters. These optimum number of clusters is used for assigning ratings. The categorized weightages and computed ratings are used to calculate the drought hazard for every region as below.

$$DH = \sum_{i=1}^{t} weights_i \times ratings_i \qquad (5)$$

where $t$ is the length of MSDI time series. Although the weightages and ratings are intrinsically linked, the above scheme assures drought hazard quantification based on magnitudes and frequencies. The $DH$ values from Eq 5 are standardized as shown below to obtain $DHI$ that varies between 0 and 1.

$$DHI = \frac{DH - DH_{min}}{DH_{max} - DH_{min}} \qquad (6)$$

The weighing and rating scheme to calculate DHI for a randomly chosen grid are given in Table S2.

**2.2.2 Drought vulnerability assessment**

Drought vulnerability forms another important component of drought risk assessment. Several aggregation techniques have been employed in the past studies to combine the drought vulnerability indicators to assess drought vulnerability. However, we use the robust method – TOPSIS (Hwang and Yoon, 1981) owing to its lesser rank reversal probabilities (Sahana et al., 2021). The steps involved in drought vulnerability assessment is outlined below.

1. Standardization of numerical drought vulnerability indicators (irrigation index, water body fraction, groundwater availability, population density and GDP) is carried out such that their values vary between 0 and 1.

$$Std. Indicator = \frac{Indicator - Indicator_{min}}{Indicator_{max} - Indicator_{min}} \qquad (7)$$

Suitable weights are assigned to categorical drought vulnerability indicators (LULC, slope and soil texture), following Thomas et al. (2016) and Sahana et al. (2021) (Table S1). This gives the decision matrix $n_{ij}$, where $i = 1,2, \dots n$ represents the number of regions and $j = 1,2, \dots m$ represents the number of drought vulnerability indicators.

2. The above decision matrix $n_{ij}$ is associated with the indicator weights $w_j$ obtained from the Analytic Hierarchy Process (AHP) method (Sahana et al., 2021). This gives the weighted decision matrix $v_{ij}$

$$v_{ij} = w_j n_{ij} \qquad (8)$$

3. Positive ($A^+$) and Negative ($A^-$) Ideal solution is calculated for each of the indicators.

$$A^+ = (v_1^+, v_2^+, \dots v_m^+) = \left[\left(\max v_{ij} | j \in I\right), \left(\min v_{ij} | j \in J\right)\right] \qquad (9)$$

$$A^- = (v_1^-, v_2^-, \dots v_m^-) = \left[\left(\min v_{ij} | j \in I\right), \left(\max v_{ij} | j \in J\right)\right] \qquad (10)$$

where $I$ and $J$ are associated with the benefit and cost criteria respectively. Here population density, LULC, slope and soil texture that bear positive correlation with the drought vulnerability are considered as benefit criteria. On the other hand, irrigation index, groundwater availability, waterbody fraction and GDP that bear negative correlation with drought vulnerability are considered as cost criteria.

4. Positive ($d_i^+$) and negative ($d_i^-$) separation measures for each region $i$ are computed based on $A^+$ and $A^-$ (also shown in Figure 1)

$$d_i^+ = \sqrt{\sum_{j=1}^{m}(v_{ij} - v_j^+)^2} \tag{11}$$

$$d_i^- = \sqrt{\sum_{j=1}^{m}(v_{ij} - v_j^-)^2} \tag{12}$$

5. Relative closeness ($R_i$) of each region to the Positive Ideal Solution is calculated as

$$R_i = \frac{d_i^-}{d_i^- + d_i^+} \tag{13}$$

$R_i$ signifies vulnerability of region $i$ to drought. $R$ is further standardised to vary between 0 and 1 to obtain drought vulnerability index (DVI)

$$DVI = \frac{R - R_{min}}{R_{max} - R_{min}} \tag{14}$$

**2.2.3 Drought risk assessment**

The hazard and vulnerability information computed in the form of DHI and DVI respectively, are combined to evaluate the drought risk. Accordingly, the drought hazard capturing the droughts in baseline (1980-2015), Near (2021-2060) and Far (2061-2099) future period is combined with drought vulnerability at 2010, 2060 and 2099 respectively. The definition of risk as provided by IPCC (AR5) (IPCC, 2014) is adopted. Though the AR5 delineates exposure as separate component of the risk, we have included exposure to be an integral part of the vulnerability following Vittal et al. (2020), since such a definition is unlikely to affect the overall conclusions of risk assessment.

$$Risk = f(Hazard, Vulnerability) = DHI \times DVI \tag{15}$$

Drought risk values computed using Eq 15 are further standardized spatially to obtain the Drought Risk Index (DRI). Standardization of drought risk at each grid is carried out using the equation

$$DRI = \frac{Rsik - Risk_{min}}{Risk_{max} - Risk_{min}} \tag{16}$$

Standardization is performed such that the values are distributed between 0 and 1, so as to classify different risk categories. Further, circumstances such as highly vulnerable population being exposed to mild droughts or no droughts at all may arrive, and are handled well due to the integrated assessment of drought risk. For eg., if the hazard is low in a region, it is likely to be classified as 'low to moderate' in terms of drought risk, despite having high vulnerability.

Apart from representing the risk as product of hazard and vulnerability, it can also be represented using bivariate choropleth (Mohanty et al., 2020). Colorscale of these bivariate choropleth is characterized by all possible combinations of DHI and DVI classes. Such maps clearly demarcate the hazard-driven and vulnerability-driven risk.







**Table 1. Drought vulnerability indicators used for drought vulnerability assessment. The sources for indicators in baseline period and projected period along with their relevance and correlation with drought vulnerability is presented. Representative indicators to arrive at the drought vulnerability indicators for projected period are also listed.**


| Data | Relevance to drought vulnerability | Correlation with drought vulnerability | Past studies using this data | Observed | | | | | Projected | |
|---|---|---|---|---|---|---|---|---|---|---|
| | | | | Source | Period | Spatial resolution | Units | Details | Source | Representative Indicator |
| **Population density** | Demographic attribute for assessing social vulnerability and exposure. | Positive | (Carrão et al., 2016; Rajsekhar et al., 2015) | NASA Socioeconomic Data and Applications Centre (SEDAC) (http://sedac.ciesin.columbia.edu/data/set/gpw-v4-population-density) | 2010 | 1 km | person/km² | • Population density estimates are based on the national censuses and population registers. <br>• Given as population count by area. | Inter-Sectoral Impact Model Intercomparison Project (ISIMIP2b experiments) data archive (Warszawski et al., 2014) | Population (SSP2) |
| **GDP** | Economic welfare for assessing economic vulnerability as well as adaptive capacity. | Negative | (Carrão et al., 2016; Naumann et al., 2014; Wu et al., 2017) | (Ghosh et al., 2010) | 2006 | 1 km | millions of dollars | • Defense Meteorological Satellite Program's Operational Linescan System (DMSP-OLS) nighttime imagery by NOOA to calculate total GDP (Ghosh et al., 2010) | | GDP (SSP2) |
| **Irrigation Index** | Adaptive capacity component. High irrigation ratio implies high adaptive capacity and lower drought vulnerability. | Negative | (Murthy et al., 2015; Wu et al., 2017) | Web based land use statistics information system https://aps.dac.gov.in/LUS/Index.htm | 2010 | District | - | • Data published by Directorate of Economics & Statistics, Department Agriculture, Cooperation & Farmers Welfare. <br>• Land use statistics information system is designed and developed by Agriculture Informatics Division, National Informatics Centre, Ministry of Communication & IT, Govt. of India, New Delhi. <br>• Given as the ratio of irrigated area to cropped area. | | Irrigation water consumption, Irrigation water withdrawal (RCP2.6 –SSP2 & RCP6.0-SSP2) |
| **Water bodies fraction** | Water resources (streams/rivers) and water infrastructure (dams/reservoirs) for assessing the physical vulnerability, and provides adaptive capacity. | Negative | (Naumann et al., 2014) | Bhuvan-Indian Geo Platform | 2010 | 3' | - | • Advanced Wide Field Sensor (AWiFS) satellite imagery is used by NRSC to extract the water bodies fraction. | | Surface runoff, Total runoff, Total water storage (RCP2.6 –SSP2 & RCP6.0-SSP2) |
| **Groundwater** | Adaptive capacity component to cope with drought. | Negative | (Pandey et al., 2010) | Dynamic Ground Water Resources of India, Central Ground Water Board Ministry of Water Resources, Report on July 2011, (CGWB, 2014) | 2011 | District | ham | • Groundwater resources assessment based on the State and Central groundwater boards of India. <br>• Net groundwater availability estimates are based on the annual replenishable groundwater resources and the natural discharge during non-monsoon season. | | Groundwater runoff, Total water storage (RCP2.6 –SSP2 & RCP6.0-SSP2) |
| **Land Use Land Cover (LULC)** | Accounts for social vulnerability to drought due to exposure. | Positive | (Pandey et al., 2010; Thomas et al., 2016) | The USGS Land Cover Institute (LCI) (https://landcover.usgs.gov/global_climatology.php) | 2001-2010 | 0.5 km | - | • The Collection 5.1 Moderate Resolution Imaging Spectroradiometer (MODIS) Land Cover Type (MCD12Q1) product for the period 2001-2010 issued by Broxton et al. (2014) to develop global land cover. | NASA Earthdata from ORNL DAAC (Chini et al., 2014) (https://doi.org/10.3334/ORNLDAAC/1248) | Fractional Land Use Land Cover data (RCP2.6 & RCP6.0) |
| **Digital Elevation Model (DEM) (Slope)** | Spare time for water retention bestows higher adaptive capacity in flat slope parts. Accounts for physical vulnerability to drought. | Positive | (Ekrami et al., 2016; Pandey et al., 2010) | SRTM 90 m Digital Elevation Database v4.1 (http://www.cgiar-csi.org/data/srtm-90m-digital-elevation-database-v4-1#download) | 2007 | 90 m | m | • NASA Shuttle Radar Topography Mission elevation data derived from interferometric techniques. | | Constant (same as observed) |
| **Soil Type** | Water holding capacity of soil based on the textural properties. Accounts for social vulnerability to drought due to exposure. | Positive | (Pandey et al., 2010; Thomas et al., 2016) | FAO Harmonized World Soil Database (HWSD) (http://www.fao.org/soils-portal/soil-survey/soil-maps-and-databases/harmonized-world-soil-database-v12/en/) | 2003 | 1 km | - | • Major contributors of the soil data for the Indian regions are All India Soil and Land use Survey (1965) and the International soil map of vegetation by India Council of Agricultural Research (FAO-UNESCO, 1977). <br>• Loamy soils are more vulnerable to drought compared to clayey soils. | | |

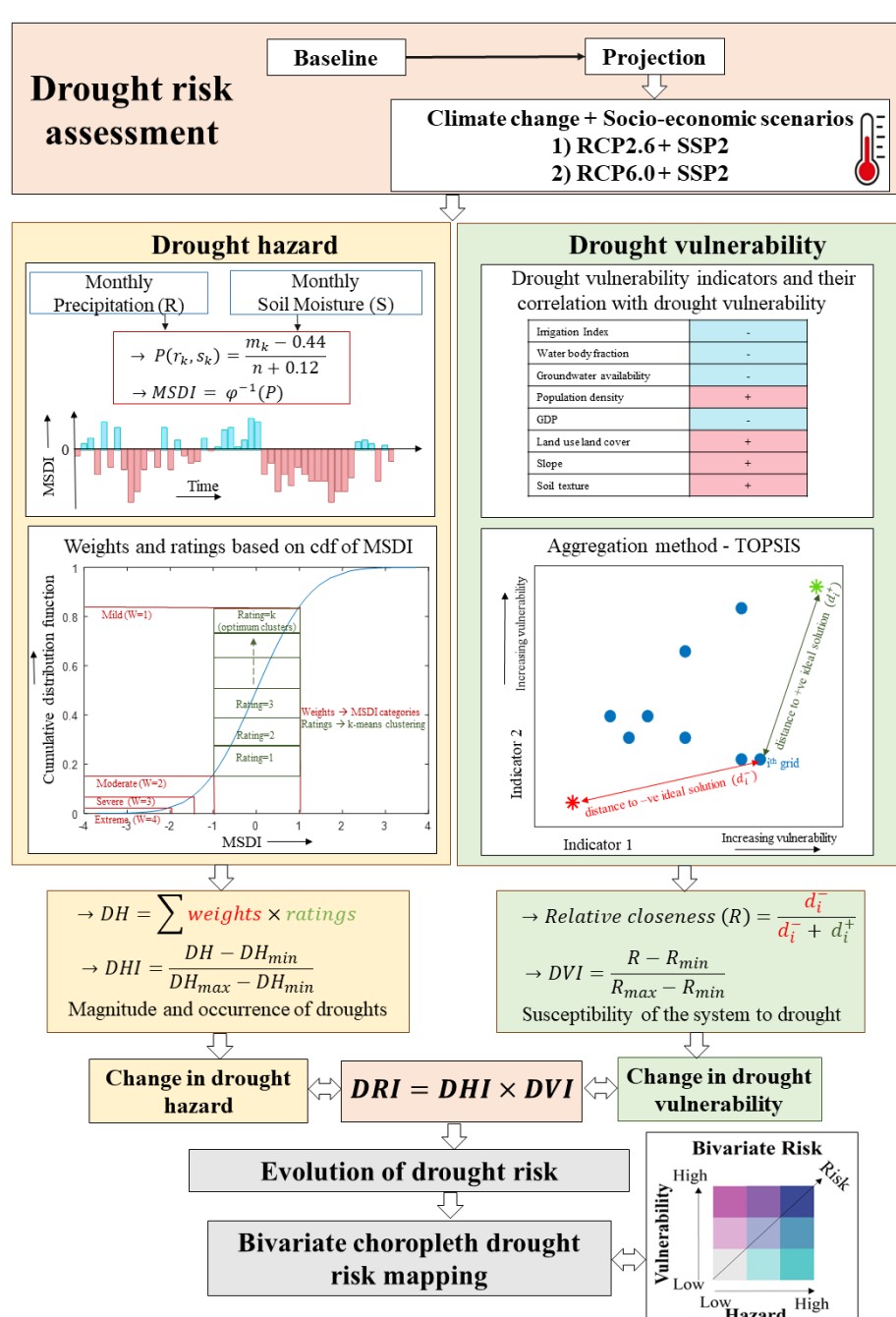

**Figure 1. Framework to assess drought risk evolution. Monthly rainfall and monthly soil moisture is used to compute multivariate standardized drought index (MSDI). Weights and ratings system of MSDI is adopted to further compute drought hazard index (DHI). Multi-criteria decision making technique – TOPSIS is used to calculate drought vulnerability index (DVI) considering eight drought vulnerability indicators. The product of DHI and DVI is the drought risk index (DRI). Drought risk assessment is carried out for Baseline period (1980-2015), Near future (2021-2050) and Far future (2061-2100) for various climate and socio-economic scenarios.**


## 3. Results and discussion

### 3.1 Drought hazard

#### 3.1.1 Projection of hydro-climatic variables

The multi-model ensemble precipitation and soil moisture data from the four GCMs is used for drought hazard assessment. The country-wide accumulated data (summed over all grids) of these hydro-climatic variables is shown in Figure 2. The projected precipitation as well as soil moisture for the RCP6.0 scenario is high compared to the RCP2.6 scenario. Further, it is noted that the variability in both the variables increases with time. However, the variability in the hydro-climatic variables in the baseline period is high compared to the projected period.

#### 3.1.2 Projection of drought hazard

The multi-model ensemble drought hazard for different RCP scenarios and time slices along with the baseline period are shown in Figure 3. The indices representing drought hazard are classified into five categories based on equal classification scheme: 0-0.2 (very low), 0.2-0.4 (low), 0.4-0.6 (medium), 0.6-0.8 (high) and 0.8-1 (very high). The MSDI-based drought hazard maps developed for the baseline period matches well with hazard maps developed from other multivariate indices such as SPEI (Gupta et al., 2020), as compared to those developed from the univariate SPI (Vittal et al., 2020). It is observed that the projected hazard over many regions is less severe compared to the baseline period. However, certain parts of north-western India and east coastal regions are under high drought hazard class. The hazard transition from the baseline to different scenarios is presented in Figure 4. The baseline and projected scenarios of drought hazard are represented using five different classes – very low, low, medium, high and very high. Every region (grid) of the country may transit from one class in the baseline scenario to another class in the projected scenario, or remain in the same class for both baseline and projected scenario. In the transition matrix we compute the % area of the country that transitioned from one hazard class to other, to quantify the effect of climate change. The upper triangle in the figure represents % area transition from lower to higher hazard classes, the lower triangle represents % area transition from higher to lower hazard classes, and the diagonal elements represent % area with no transition.

In general, a transition from higher hazard classes to lower hazard classes is observed under the projected scenarios, implying that more regions in the country are expected to come under low hazard category in the future. From Figure 2a, S2 and S3, we see that precipitation and soil moisture for the projected period show an increasing trend. Further, it is to be noted that the hazard assessment using MSDI is based on the long term mean and variability of these drought indicators under a probabilistic analysis framework, and not necessarily the magnitudes of precipitation and soil moisture. Here we see that the projections of these indicators exhibit lower variability compared to the baseline period (Figure 2a). Therefore, it is observed that many regions undergo transition from high hazard to low hazard. The future drought hazard assessment using the projected hydro-climatic variables revealed that more than 35% area of the country is expected to be under the low hazard class, as compared

to 8% in the baseline period (refer Figure 4 and Figure 10). It is also interesting that the area under high hazard class is greater in the Far future as compared to the Near future irrespective of the RCP scenarios. This is ascribed to the higher variability of the hydro-climatic variables in Far future compared to the Near future period that resulted in higher magnitude of drought events. Of all the future drought hazard scenarios considered, the RCP2.6-Far scenario revealed the largest area (2.8%) under high and very high hazard classes. This accounts for a 7% reduction in high and very high hazard classes compared to the

baseline scenario. It is observed that North-Western India, parts of Jammu, Kashmir, Andhra Pradesh and Marathwada come under high hazard classes.

It is interesting to note that the probabilistic Budyko framework-based projected annual per capita water availability analysis (PCWA) for the Indian region by Singh & Kumar (2019) show a decrease in PCWA in 2.0 °C warmer world compared to 1.5 °C warmer world under CMIP5-based mitigation, medium stabilization and high-end (RCP8.5) climate change scenarios,

indicating high hazard in the Far future. Similarly, higher drought hazard is observed in the Far future compared to the Near future by Gupta & Jain (2018) & Gupta et al. (2020), who performed SPEI-based drought hazard analysis using CMIP5 GCMs under high-end climate change. Further, frequency-based soil moisture droughts analysis by Aadhar & Mishra (2020, 2021) and SPEI-based drought frequency analysis by Zhai et al. (2020) show an increased drought frequency in the future period over South Asia compared to the baseline period. This shows that Far future period is more prone to drought hazard than the

Near future. On the other hand, few studies such as Koutroulis et al. (2019) & Cook et al. (2020), who used CMIP5 and CMIP6 simulations respectively show that drought exposure/frequency over the Indian region decrease with time. Such contradicting observations are possibly due to selection of low-skill GCMs (Aadhar and Mishra, 2020), in Koutroulis et al. (2019) & Cook et al. (2020). It is to be noted that the four GCMs considered in the present study for precipitation and soil moisture simulations are bias-corrected for precipitation, and covers more uncertainty in temperature and precipitation changes compared to other

GCM subsets (McSweeney et al., 2015). However, inclusion of other skilled GCMs can account for wide range of uncertainty in the drought hazard assessment.

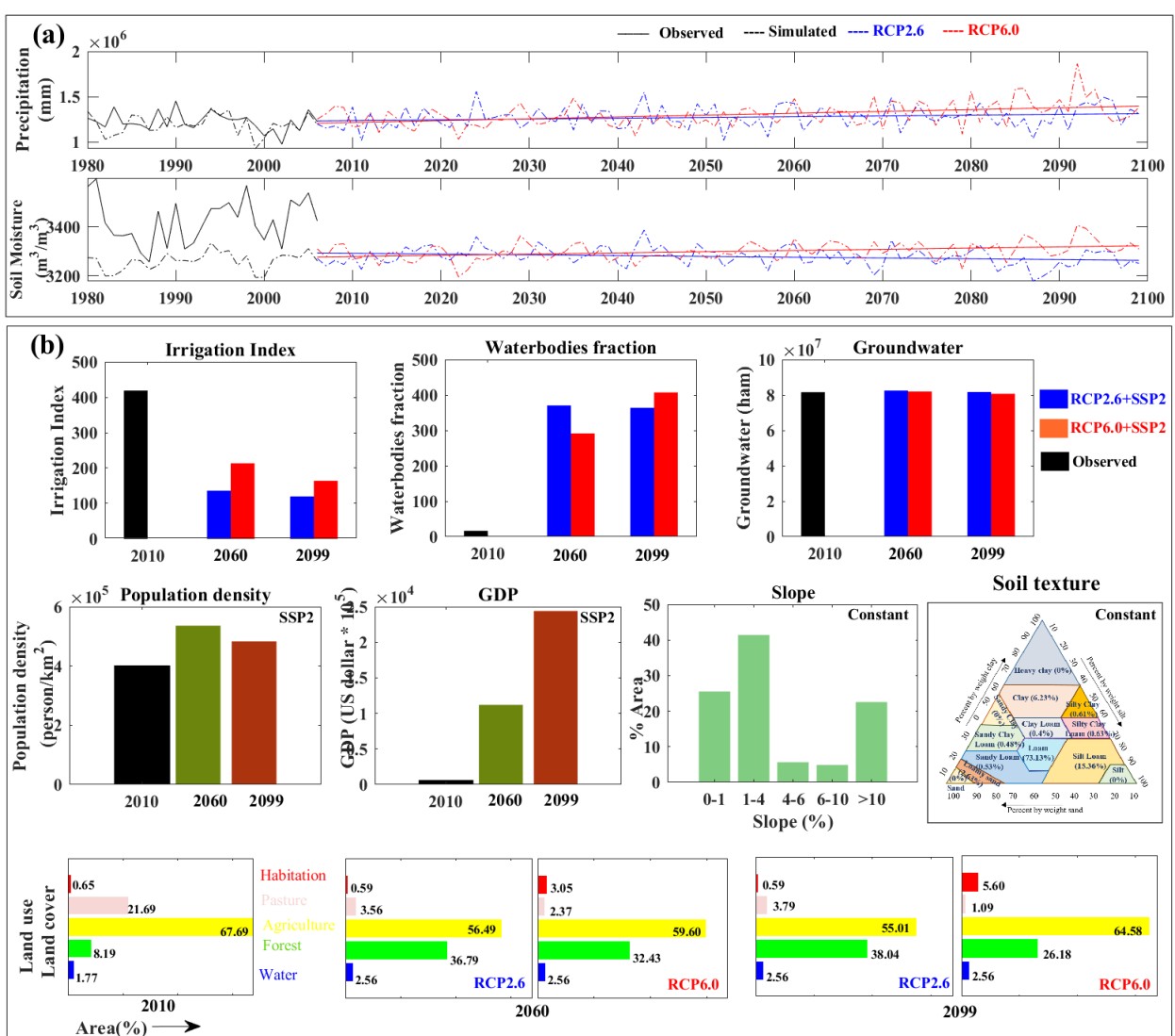

**Figure 2. Datasets used for drought risk assessment. a) Projected hydro-climatic variables such as monthly precipitation and monthly soil moisture are used for drought hazard assessment. b) Projected drought vulnerability indicators such as irrigation index, water body fraction, groundwater availability, population, GDP and land use land cover, along with static drought vulnerability indicators such as slope and soil texture are used for drought vulnerability assessment. Datasets for projected period are divided into Near future (2021-2060) and Far future (2061-2100) to check the evolution of drought risk.**

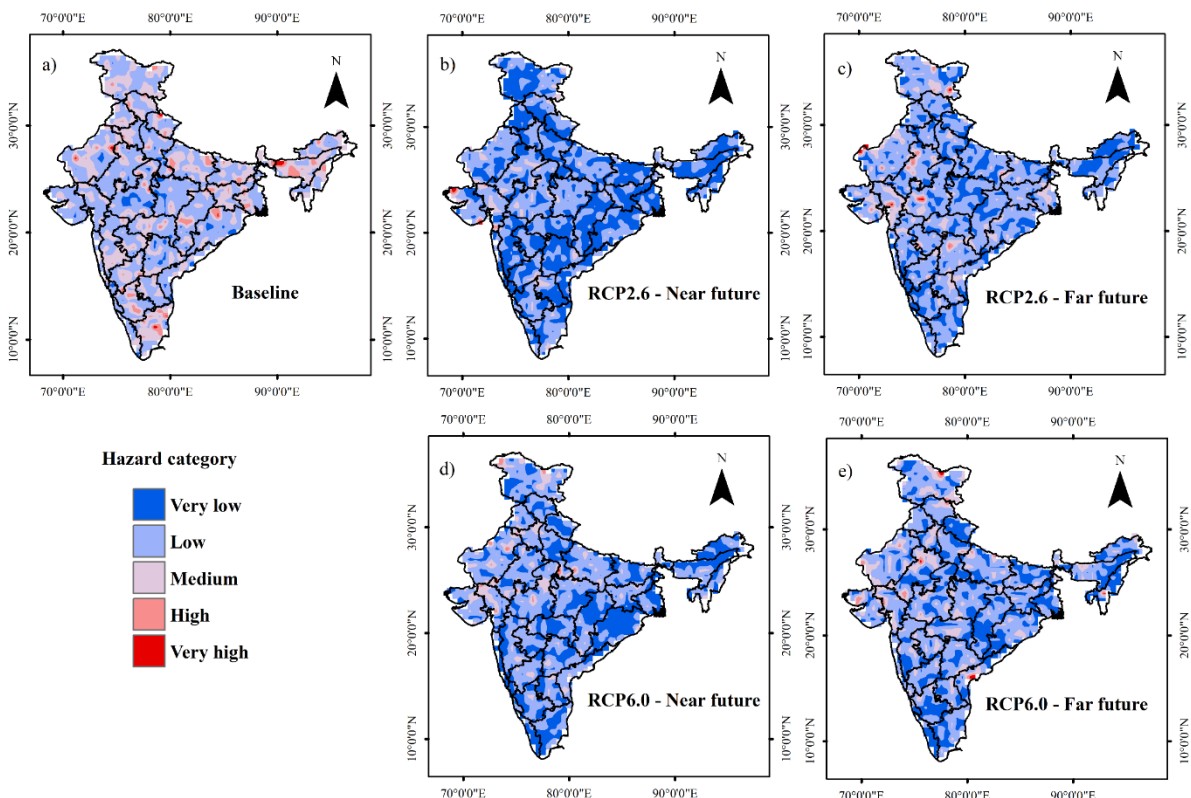

**Figure 3. Multi-model ensemble drought hazard maps for the scenarios a) Baseline, b) RCP2.6 Near future, c) RCP2.6 Far future, d) RCP6.0 Near future, e) RCP6.0 Far future.**

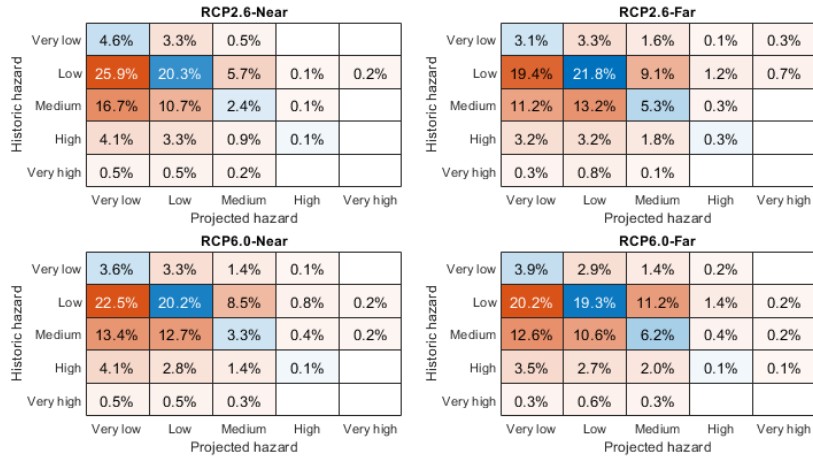

**Figure 4. Transition of drought hazard from baseline period to projected period. The value in each cell represents the change in % area of the country from one hazard class to another. Red color shows transition, and blue represents no transition.**


### 3.2 Drought vulnerability

#### 3.2.1 Projection of drought vulnerability indicators

The varying drought vulnerability indicators for the drought vulnerability assessment is shown in Figure 2. It is observed that GDP increases with time continuously, whereas population reaches its peak during the end of Near future (2060) and decreases gradually by the end of the century. The representative indicators obtained through human influences, varying land use and water abstractions according to the RCP2.6-SSP2 and RCP6.0-SSP2 conditions are used to derive the drought vulnerability indicators such as irrigation index, waterbody fraction and groundwater availability for the projected period. It is observed that the irrigation index decrease with time for RCP2.6-SSP2 and RCP6.0-SSP2 projections. Water body fraction remains constant for RCP2.6-SSP2 projection and increases with time for RCP6.0-SSP2 projection. Further, groundwater availability remains constant for RCP2.6-SSP2 and RCP6.0-SSP2 projections. The biggest difference in land use land cover changes is observed in RCP6.0 condition compared to RCP2.6. It is also seen that % area under habitation increases continuously with time in the case of RCP6.0. Slope and soil texture data is assumed to be constant (Figure S7).

#### 3.2.2 Projection of drought vulnerability

The multi-model ensemble drought vulnerability projections for different scenarios is presented in Figure 5. It is observed that many regions of the country are expected to be more vulnerable to drought compared to the baseline period. In general, parts of North-Western, eastern India and southern coast are observed to be under high vulnerability class in the future scenarios. The transition of drought vulnerability from one class of vulnerability from baseline to another class of vulnerability in the future is given in Figure 6. It can be observed that the drought vulnerability under RCP6.0-SSP2 scenario is worst compared to the RCP2.6-SSP2 scenario, since high transition from lower vulnerability classes to higher vulnerability classes is observed in the former case. As high as 42.9% area transits from lower vulnerability classes to higher vulnerability classes under RCP6.0-SSP2 Near future. Also, a 33% increase in the area under high and very high vulnerability classes is observed in this worst-case scenario, with North-Western India, Western Coast and parts of Chattisgarh, Odisha and Jharkhand under very high vulnerability class.

In the global freshwater vulnerability analysis conducted by Koutroulis et al. (2019), although they show that sensitivity component of the overall freshwater vulnerability is increasing with time, an increasing adaptive capacity and decreasing exposure is reducing India's vulnerability to drought. However, our study shows an increasing vulnerability to drought, considering sensitivity, adaptive capacity as well as exposure factors. Such contradicting observations in drought vulnerability is possibly due to the choice of low-skill GCMs in Koutroulis et al. (2019). Further, the socio-economic challenges for adaptation and mitigation in different SSP narratives are lead by different development pathways (O'Neill et al., 2017). Therefore, adoption of other SSPs in drought vulnerability assessment may unveil other plausible drought vulnerability projections.

Next, we aggregate hazard and vulnerability information on meteorological sub-division scale (Meteorological sub-divisions are the meteorologically homogenous regions identified by India Meteorological Department (Kelkar and Sreejith, 2020)) to

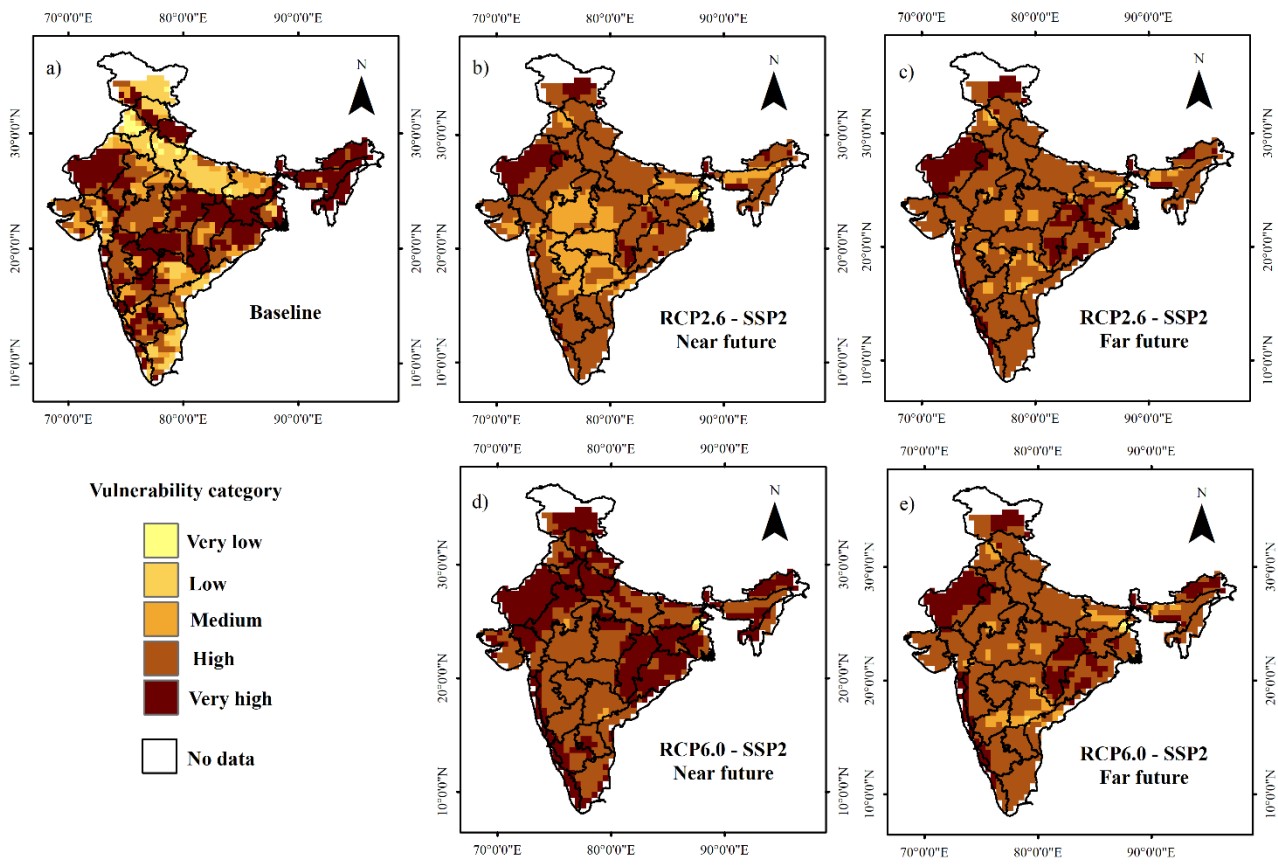

**Figure 5. Multi-model ensemble drought vulnerability maps for the scenarios a) Baseline, b) RCP2.6-SSP2 Near future, c) RCP2.6-SSP2 Far future, d) RCP6.0-SSP2 Near future, e) RCP6.0-SSP2 Far future.**

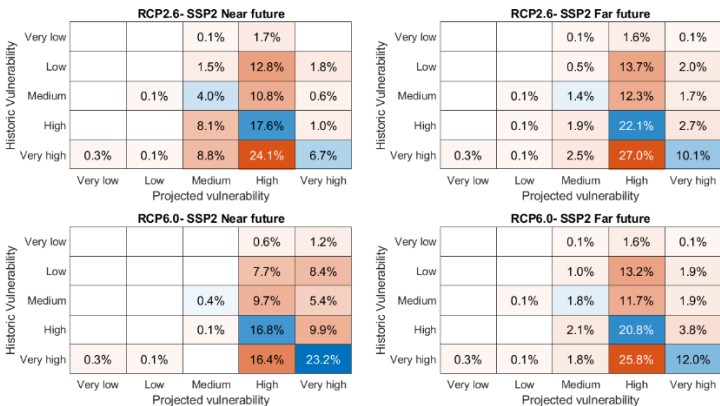

**Figure 6. Transition of drought vulnerability from baseline period to projected period. The value in each cell represents the change in % area of the country from one vulnerability class to another. Red color shows transition, and blue represents no transition.**

identify the sub-divisions under critical drought condition due to the interplay of hazard and vulnerability . Scatter of drought hazard and drought vulnerability for 30 sub-divisions is shown in Figure S8. It is seen that West Rajasthan, Haryana and West Uttar Pradesh sub-divisions are expected to have high drought risk compared to the other sub-divisions in all the scenarios. Further, the number of sub-divisions falling under critical drought risk (DHI > 0.25, DVI > 0.75) is high in the case of RCP6.0-SSP2 scenario, with 22 meteorological subdivisions under high vulnerability (DVI > 0.75), particularly in RCP6.0-SSP2-Near future scenario.

## 3.3 Drought risk

### 3.3.1 Projection of drought risk

The multi-model ensemble drought hazard and vulnerability projections under different scenarios are combined according to Eq 15 to obtain drought risk projections (Figure 7). It is to be noted that the validation of drought risk map for the baseline period (1980-2015) has been carried out by Sahana et al. (2021), based on the disaster data in terms of number of people affected. It is noted that parts of Rajasthan, Madhya Pradesh, Maharashtra, Orissa and Tamil Nadu, Kerala, Chattisgarh, Haryana, Himachal Pradesh, Chandigarh, Assam and Nagaland that are under moderate to severe drought risk category, have experienced moderate to worst drought disaster. Further, the drought risk estimates for the baseline period from the present study compares well with regional-scale drought risk studies in India such as those for Andhra Pradesh (Murthy et al., 2015), Bearma basin (Thomas et al., 2016), Maharashtra (Swami and Parthasarathy, 2021). From the drought risk projections, it is noted that parts of the North-Western India is expected to be more prone to drought risk compared to the baseline period. On the other hand, Central Indian regions are expected to switch to lower risk classes. The transition of drought risk from one class of vulnerability from baseline to another class of risk in the future is given in Figure 8. Highest transition (30% area)

from lower risk to higher risk classes is observed in RCP6.0-SSP2-Far future scenario. Also, overall drought risk reduces by 0.8% in this scenario compared to the baseline. It is interesting to note that the RCP6.0-SSP2 Far future scenario is not the worst-case scenario in drought vulnerability projection, yet it turned out to be worst-case scenario in drought risk projection due to high drought hazard projection, revealing the importance of comprehensive drought risk assessment. Risk is an outcome of interaction between hazard and vulnerability, and is also a function of time. The fact that worst case scenarios are different for drought hazard and drought vulnerability, indicates dissimilar behavior of drought hazard and vulnerability indicators in inducing drought risk. For eg. population density is high in the Near future period (2060) as compared to the Far future (2100), while precipitation is continuously increasing in the projected period. A combination of such different hazard and vulnerability behavior in a given time period is effectively captured through comprehensive risk analysis. Therefore, though RCP6.0-SSP2 Far future scenario is not the worst-case scenario for drought vulnerability compared to RCP6.0-SSP2 Near future, interaction of high hazard with moderate to high vulnerability resulted in worst drought risk scenario in the case of RCP6.0-SSP2 Far future. However, in general, when the change in drought risk for all the future scenarios are compared with the baseline, it is observed that area falling under drought risk due to drought vulnerability is increased (Figure 9). It is to be noted that the water availability projections for India by Koutroulis et al. (2019) show decreasing drought risk with time, as opposed to the increasing drought risk from the present study. The choice of climate change scenarios and climate models by Koutroulis et al. (2019) could be a possible reason for such difference. Further, projected bivariate choropleth maps for unique combinations of DHI and DVI is presented in Figure 9. It is seen that most of the regions are constituted by low hazard and high vulnerability indicating high impact of societal developments rather than climate-invoked changes. Hence it is important to take the drought mitigation plans based on the socio-economic conditions instead of just considering hydro-climatic conditions of the interested region. Consolidated results showing the % area of different classes of drought hazard, vulnerability and risk under various climate and socio-economic scenarios are given in Figure 10. Of all the future drought hazard scenarios considered, the RCP2.6-Far scenario revealed the largest area (2.8%) under high and very high hazard classes. In the case of drought vulnerability, as high as 42.9% area transits from lower vulnerability classes to higher vulnerability classes under RCP6.0-SSP2 Near future, with 93% area of the country under high and very high drought vulnerability class. Further, in the worst case drought risk scenario (RCP6.0-SSP2 Far future), it is observed that 2.7% area of the country is under high and very high drought risk class.

### 3.3.2 Potential applications

The drought hazard, vulnerability and risk projection maps from the present study, developed at 0.5° lat.× 0.5° lon spatial resolution are comparable with blocks/district level area. Therefore, these maps can assist the block-level administrators to know region-specific causative factors inducing severe drought risk both in baseline and projected period, besides the indigenous components governing the drought risk. Also, these maps can inform the state or federal disaster management authorities concerning the climate action plans. The change in drought risk at different projected periods can modulate

adaptation and mitigation strategies and can be included in decision support system for drought management. Since drought risk is found to be mainly driven by societal factors, action plans should be directed to improve socio-economic conditions. Groundwater conservation, conjunctive use of surface and groundwater, farmer participation in crop insurance, water saving farm practices and technologies are some important measures that can be adopted for raising the socio-economic standards. Further, the framework of our study is applicable for state-wise drought risk assessment with reliable hydro-climatic and socio-

economic indicators. Such an assessment can recommend measures for watershed management, irrigation and agricultural practices and reorganizing water demand and supply management at a local scale.

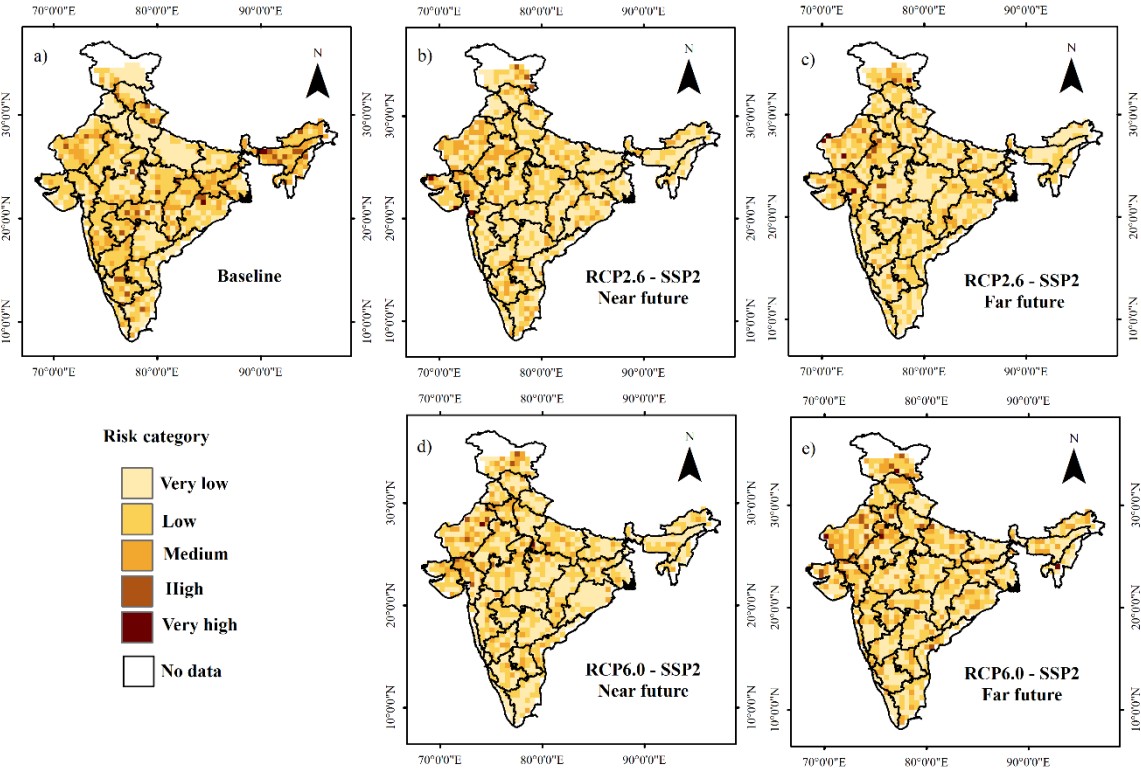

**Figure 7. Multi-model ensemble drought risk maps for the scenarios a) Baseline, b) RCP2.6-SSP2 Near future, c) RCP2.6-SSP2 Far future, d) RCP6.0-SSP2 Near future, e) RCP6.0-SSP2 Far future.**

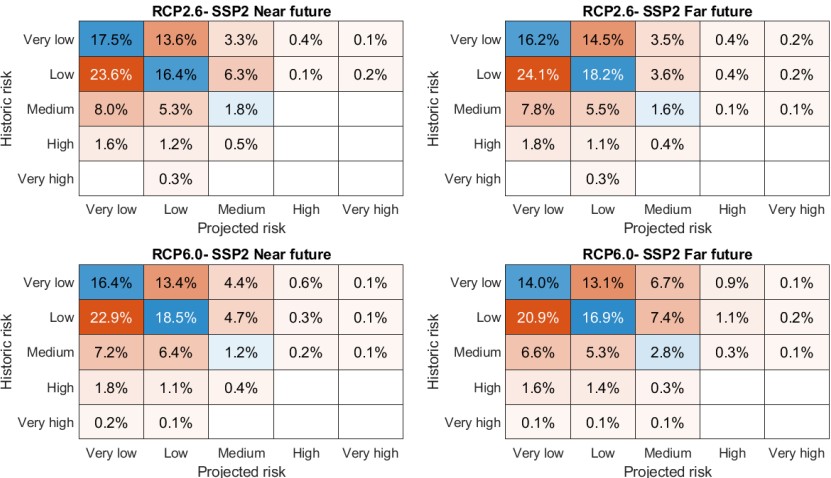

**Figure 8.** Transition of drought risk from baseline period to projected period. The value in each cell represents the change in % area of the country from one risk class to another. Red color shows transition, and blue represents no transition.

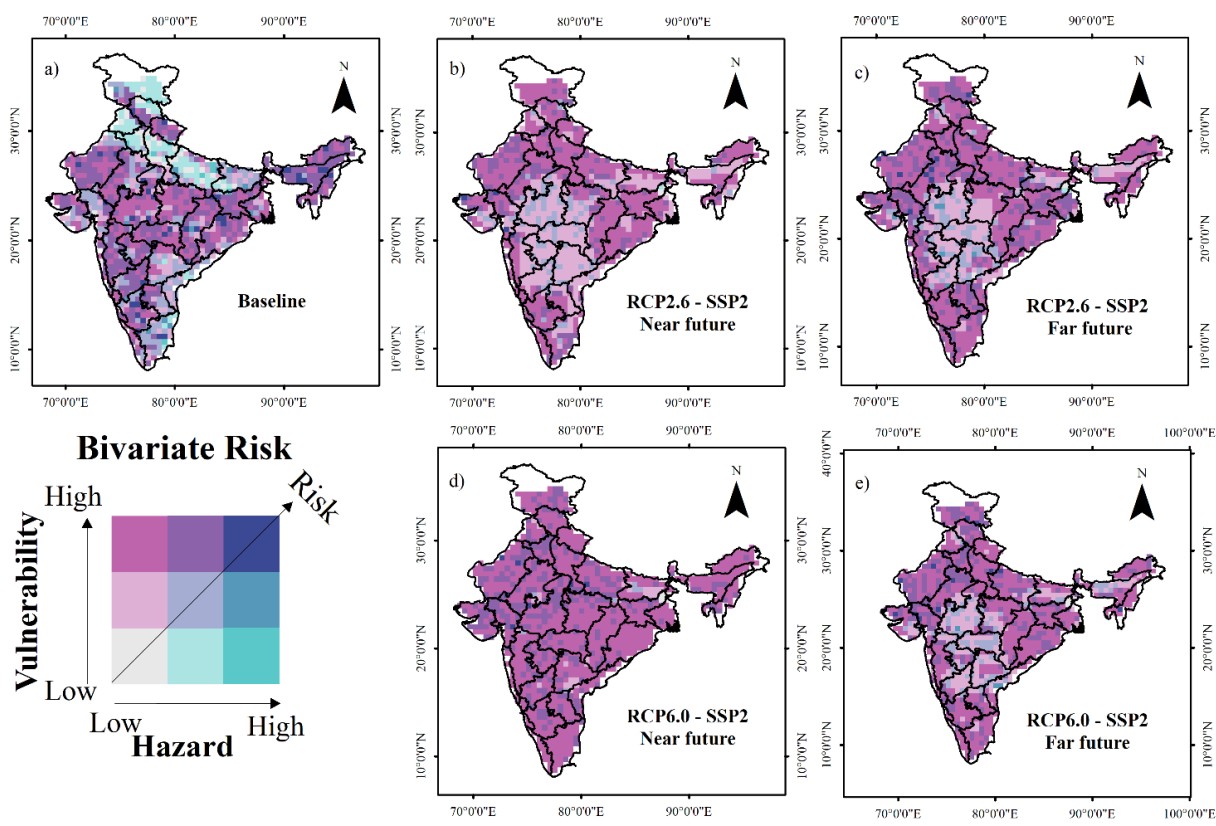

**Figure 9.** Bivariate choropleth drought risk maps showing hazard-driven and vulnerability-driven drought risk for the scenarios a) baseline, b) RCP2.6-SSP2 Near future, c) RCP2.6-SSP2 Far future, d) RCP6.0-SSP2 Near future, e) RCP6.0-SSP2 Far future.

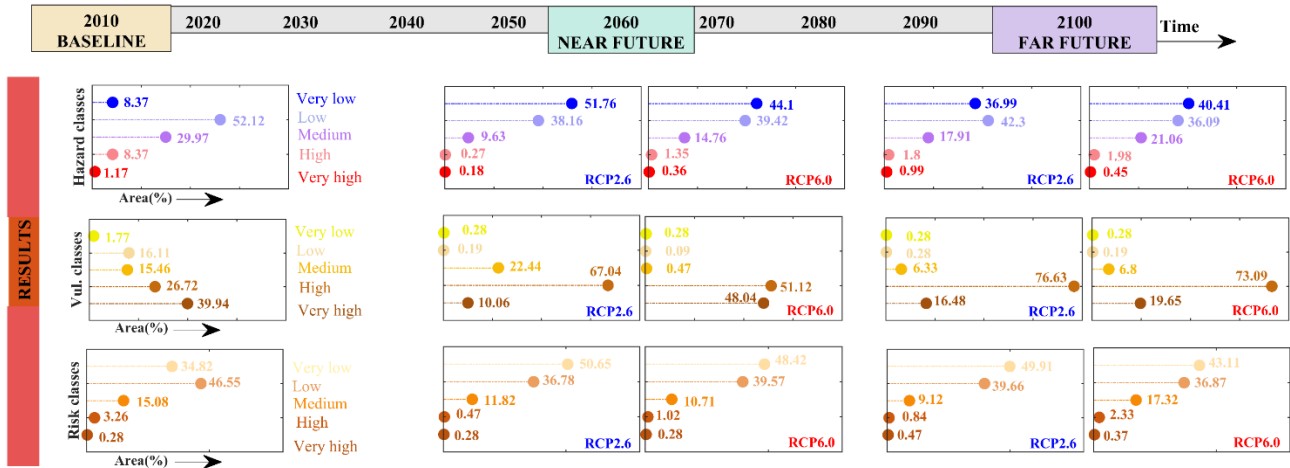

**Figure 10. Summary of drought risk evolution. % area of different classes of drought hazard, vulnerability and risk under various climate and socio-economic scenarios.**

## 4. Concluding remarks

This study presents future projections of drought risk over India under changing climate and socio-economic conditions. This is achieved combining the drought hazard and drought vulnerability projections. Drought hazard assessment is carried out using a multivariate drought index known as MSDI, an indicator of agro-meteorological drought. Drought vulnerability is assessed using a robust MCDM technique called TOPSIS, considering changes in relevant socio-economic indicators. Drought risk projection studies undertaken over the Indian region are based on drought hazard alone, and no consideration has been given to the drought vulnerability component. The present study quantifies the relative contribution of drought hazard and drought vulnerability to the overall drought risk projections under a comprehensive risk framework. Thus, our analysis can aid different stakeholders involved in drought management for adaptation and mitigation plans under changing climate and socio-economic conditions. This marks the significant improvement of our study over existing studies on drought risk assessment in India under climate change. Further, we present for the first time, future projected bivariate choropleth plots to identify the drivers of overall drought risk across the country. The multi-model ensemble drought hazard and drought vulnerability are computed for the two RCP-SSP scenarios- RCP2.6-SSP2 and RCP6.0-SSP2 for the Near and Far future timelines. The current study is limited by simulations from a single global vegetation model rather than multiple impact models including hydrologic or land surface simulations. Important conclusions of the study are outlined below.

1. The MSDI-based drought hazard assessment reveals that more than 35% area in India is projected to be under low hazard class as opposed to 8% in the baseline period, possibly due to rising precipitation in the region as projected by climate models. RCP2.6-Far scenario shows 2.8% area of the country under high and very high hazard classes,

accounting for 7% reduction in those two drought hazard categories. In general, the spatial extent of high and very high hazard classes is greater in Far future as compared to the Near future.

2. Drought vulnerability is projected to increase for all scenarios, with 77% area under high or very high vulnerability class as compared to 66% in the baseline period. A rise in 33% of area under high or very high vulnerability class is observed in RCP6.0-SSP2-Near future scenario.

3. Among the two RCP-SSP scenarios considered, RCP6.0-SSP2 scenario exhibits worst case of drought vulnerability due to high transition from lower to higher vulnerability classes as compared to RCP2.0-SSP2 scenario.

4. Integration of drought hazard and vulnerability projections shows an overall decrease in drought risk projections, resulting primarily from reduction in drought hazard. However, a transition from lower to higher risk classes ranging upto 30% is observed in RCP6.0-SSP2 future Far scenario.

5. Meteorological sub-divisions such as West Rajasthan, Haryana and West Uttar Pradesh are expected to be under high risk in the projected period under all the scenarios.

6. Bivariate choropleth analysis show that future drought risk is significantly driven by increased vulnerability resulting from societal developments rather than climate-induced changes in hazard. Therefore, future efforts on building drought resilience in the country must include strengthening socio-economic conditions.

**Author contribution**

V. Sahana and Arpita Mondal designed the study. V. Sahana conducted the analysis, generated the results and wrote the first draft. Both the authors contributed in writing and revising the manuscript.

**Competing interests**

The authors declare that they have no known competing interests or personal relationships that could have appeared to influence the work reported in this paper.

**Acknowledgements**

The authors are thankful to the IMD, Pune, NASA Modern-Era Retrospective Analysis for Research and Application – Land (https://disc.sci.gsfc.nasa.gov/), NASA Global Inventory Modelling and Mapping Studies (https://nex.nasa.gov/nex/projects/1349/) for providing essential data for drought hazard assessment, all the agencies mentioned in Table 1 for providing drought vulnerability indicator datasets and the Postdam-Institute for Climate Impact Research for providing ISIMIP data (https://esg.pik-potsdam.de/search/isimip/) for drought risk projection study. The authors appreciate the financial support received from the Science and Engineering Research Board, Department of Science and

Technology, Government of India, through the project ECR/2017/000566 and SPLICE-Climate Change Programme,

Department of Science and Technology, Government of India, through the project DST/CCP/CoE/140/2018. The authors also thank Mr. Roshan Jha for his comments on the first draft.

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
