# Peer review of "Evolution of multivariate drought hazard, vulnerability and risk in India under climate change"

_Natural Hazards and Earth System Sciences, 2022_

## Referee Comment (RC2)

**Review: Sahana & Mondal - Evolution of multivariate drought hazard, vulnerability and risk in India under climate change, NHESS**

**Summary**

This paper presents a drought risk assessment for India for a baseline period and for two RCPs and two SSPs. The methodology used seems appropriate for the data used and spatial scale considered. The authors found that drought risk was primarily comprised of the drought vulnerability component, rather than the hazard, these results were shown effectively using bi-variate maps. Overall, I found this an interesting paper with results and outcomes that could be useful for drought planning and mitigation at the high level in India. However, I found that the clarity of the paper could be improved and made more expansive making it easier to follow and reproduce elsewhere. Specific examples are discussed below. I recommend that this paper is revised before publication to clarify key methodological points highlighted below.

**Major comments**

I found the description of the methods to calculate the DHI and DVI in the supplementary information unclear, with not enough detail provided on the steps and processes with no further information provided (some examples listed below regarding weighting and standardisation). I would like to see the methods in the main body of the paper expanded. I recommend that all the whole methodology is moved to the main body of the paper, rather than the fundamental steps being in the supplementary information. I would also suggest that Figure 1 in the main body of the paper is expanded, with more detail and steps added to fully capture the methodological steps described in the paper – this is further discussed below.

Use of terminology – literature usually talks about vulnerability factors – i.e. the factors that make a person or location vulnerable to drought impacts. Also regarding vulnerability terminology, you mention components of vulnerability in the introduction (L35-38), how do the indicators (or factors) you used map onto these? This could be included in Table 1. Did you consider using for example, WorldPop data such that the vulnerability assessment could be disaggregated by sex? A final question on the factors used, how do the vulnerability factors selected address the exposure component? Although a population may be vulnerable to the impact of drought, they may not be exposed to drought or may be exposed to a lower severity of drought hazard than in other locations, for example.

The baseline period used (1980-2015) excludes the past six years, excluding significant drought events in 2016-2018 and 2021. Is there a way that updated precipitation data could be obtained and analysed to include these recent events?

L140-142: the first sentence here states that variability of the two scenarios increased over time, but the second states that the baseline period is more variable than the projected period. I am not sure how both of these statements can be true, nor am I convinced I can see these difference in the time series for precipitation or soil moisture. Please clarify this point further. This also applies to Table 3 and the text in L199-205, and Table 4 and text L232-233. Figure 7 is mentioned only on two lines at the end of Section 3.3.1, could this summary be referred to in the previous discussions of hazard, vulnerability and risk as it is a more intuitive figure to understand than tables shown in Tables 2, 3 and 4.

L233-238: Here you state that the RCP6.0-SSP2 Far future scenario is not the worst case for drought vulnerability, but was the most severe for drought risk due to the hazard component (L233-235).

Then on L236-238 in the discussion of Figure 6, you state that drought vulnerability makes up the majority of the drought risk for the same scenarios. Please could you clarify these seemingly contradictory statements. I do not disagree with the point that we need more holistic drought risk assessment though.

Section 3.3.2: At what spatial scale is this information useful to policy makers? You could consider whether the gridded data used here is relevant to that of decision makers.

L269-270: You say here that this study is an improvement for decision makers over existing drought risk assessment in India. Please briefly state what this improvement is.

**Supplementary information**

The description of the categorisation of the MSDI at the bottom of page 2 is not clear and should be expanded – for example, it should be clearly stated which categories from McKee et al., - presumably extreme drought, severe drought, moderate drought etc., but you should include the thresholds and categories used in this study here. The meaning of the following two sentences ("Further, each category is organised into sub-groups based on the occurrence probabilities of the selected category. While the weightages are assigned to MSDI categories to account for drought magnitude, ratings are assigned to the sub-groups of each MSDI category to account for drought occurrence probability") are aren't sufficiently clear; the methodology of how weights were assigned should be described more clearly – for example, you start talking about ratings and clusters, and it is not clear what these are used for.

The description of the drought vulnerability index is also overly complex and should be clarified by describing the process in words. You should also be specific for example on how exactly data were standardised and what is meant by 'suitable weights' – what makes them suitable, how have these be validated and checked?

**Minor issues**

In several places e.g. L119 and in the supplementary information you mention that data have been standardised, please explain how these data are standardised (e.g. across time or space).

L99-100: '…and brought to a monthly time-scales.' This isn't clear – how and what metric? Total precipitation? Average soil moisture? Please expand on this comment.

L139: 'the country-accumulated data of these hydro-climatic variables…' please clarify how these data were accumulated

L145: ' …many regions is less **severe** compared to the baseline period.' Word missing

L168-169: It is not clear whether this choice of low skill GCMs was in the current study or Koutroulis et al and Cook et al. – note that this same comment is also made on L209-210.

L193: Highest differences… → The biggest difference in…

L198: many region of the country **are** expected to

L211: …sub-division…

L211: Here you mention meteorological sub-divisions, what exactly does this mean, how were these defined?

L222: …expected to be under high risk to drought compared… → …expected to have a high drought risk compared…

A mix of tenses seems to be used throughout, this should be reviewed, ensuing that the past tense is used where appropriate.

In some places 'Far Future' is capitalised and others it is not – you should ensure this is consistent.

**Tables and Figures**

Table 1: Please also included the last date that the factors were updated and the date range that was available. Some of the descriptions of the datasets are unclear, for example, Water Bodies Fraction and Groundwater – it is not clear what these data actually are, and what they're measuring. It would be useful to include the unit of each factor where appropriate, e.g. for 'Groundwater', 'Water Bodies Fraction'

Table 2: I find these tables unintuitive and difficult to marry up with the description of results in the text. e.g. L149-15150: 'The future drought hazard assessment using the projected hydro-climatic variables revealed that more than 35% area of the country is expected to be under the low hazard class, as compared to 8% in the baseline period' – I can't seem to make any of the very low hazard boxes up to these numbers or the difference between them. You could consider adding a more detailed example walk through of what this table means. There is also no reference to or legend for the colour scheme used here.

Table 3 & 4: I assume these are all showing SSP2 scenarios, please clarify in the caption.

Figure 1: it would be helpful to refer to this figure in the description of methods included in the Supplementary Information. The DHI and DVI should be directly referenced in this figure to clarify when these are calculated in the processing chain – including mention of the baseline period.

Figure 4: There are some parts that are white, yet white is not mentioned in the legend.

Figure 5: I recommend adding another colour to the legend here to highlight the higher risk areas (such as a dark red or similar). Do the classes used represent any specific categories of risk?

---

## Author Comment (AC1)

**Responses and Actions taken on Reviewers' Comments**

Journal: Natural Hazards and Earth System Sciences

**Manuscript Reference No.: nhess-2022-18**

**Title**: Evolution of multivariate drought hazard, vulnerability and risk in India under climate change.

**Authors: Venkataswamy Sahana, Arpita Mondal**

We thank the Reviewer for reviewing our manuscript and providing valuable feedback that have helped improve the quality of the work significantly. In this document, we provide a point by point response and actions taken on the comments and suggestions from the reviewers.

**Responses to comments from Referee #1**

I have read your manuscript and think it covers an interesting topic. It is clearly the result of a major research effort. Including vulnerability in drought risk analysis is a known challenge, and I agree with you that looking at multiple physical drivers as well as at transient vulnerability are important steps for holistic drought risk assessments. The aim of the study is clearly stated and results are described in detail. However, the research is quite complex and so I think an extra effort is needed to make in understandable for readers of NHESS. I see some conceptual issues, but they may have been caused by a lack of understanding of the method due to its incomplete or undetailed description. In general, I think more of the method could be in the main manuscript and more details to the method (currently lacking) can be described in the supplementary material. Below, I will elaborate on the main points that I think can help improve/clarify the manuscript. In addition, I think the manuscript would benefit from a review by an English language editor, as there are multiple grammar mistakes in the manuscript and I see various possibilities for vocabulary improvements.

We thank the reviewer for the positive and constructive feedback on our work. We have now provided more details and description about the methods and they will be included in the revised manuscript and the revised supplementary material. Further, we have proof-read the manuscript and corrected for grammar and language wherever necessary. We have addressed the comments provided by you in the below sections.

I haven't listed all grammar / vocabulary mistakes, but here are a few examples from the abstract:

**e.g. L7 "a" major threat**

e.g. L10 This study investigates and evaluates the change in projected drought risk under future climatic and socio-economic conditions by combining vulnerability and hazard information at a country-wide scale for future climatic and socio-economic conditions

e.g. L18 "are found to be high risk under all scenarios"

We will consider all the above suggestions from the reviewer and revise them in the manuscript.

e.g. L15-17: Sentence is too long, it is unclear what is meant with "worst-case" scenario

We have now simplified the sentence as "In the worst-case scenario for drought hazard (RCP2.6-Far future), there is a projected decrease in the area under high or very high drought hazard classes in the country by approximately 7%. Further, the worst-case scenario for drought vulnerability (RCP6.0-SSP2-Near future) shows a 33% rise in the areal extent of high or very high drought vulnerability classes."

I think maybe "The West Utter Pradesh, Haryana, ...., regions" are meant rather than "regions of West Utter Pradesh,..."

The sentence will be rewritten as "The West Uttar Pradesh, Haryana, West Rajasthan and Odisha regions are found to be high risk under all scenarios."

In general, in the manuscript there are many sentences that are difficult to understand (too long and/or with too complex structure).

We will take care of the complex sentences and simplify them in the revised manuscript.

Below, I add some general comments and questions structured following the study aims, highlighting the most pressing questions with respect to the method.

1. Multivariate drought hazard projection using Multivariate Standardized Drought Index (MSDI) that considers concurrent deficits in precipitation and soil moisture for future warming scenarios.

a) L81: "However, droughts can often manifest as a complex interplay of multiple influencing variables necessitating a multivariate approach for characterization of drought hazard (Sahana et al., 2020). For the agrarian country of India, agro-meteorological drought hazard projections should consider deficits in precipitation or soil moisture or both" I agree looking only at PR is too narrow. It is indeed interesting to look at both, but as far as I understand the

method, only events with both a SM-deficit and a PR-deficit are considered. Is this approach justified? I can think of cases where a SM-deficit alone is enough to cause a drought impact – I feel the hazard method does not sufficiently take into account the propagation of drought through the hydrological cycle, which involves attenuation and lag effects. The manuscript displays different results than other papers: how can it be evidenced that the presented method is better and the results are more reliable than those of other studies?

We would like to clarify that the Multivariate Standardized Drought Index (MSDI) is equally capable of capturing deficits individually in precipitation or soil moisture, or their joint deficit, considering dependence between these two variables. This is a unique advantage of MSDI (Hao and AghaKouchak, 2014) over other univariate indices. This clarification will be included in the revised manuscript. Further, we have added Supplementary Figure S5 (given below) that shows how the MSDI is capable of representing the onset, propagation and termination of drought. In this figure, considering -0.8 as the threshold for drought trigger, it seen that whenever either the SPI or the SSI falls below this threshold, MSDI covers the critical trajectory and offers a conservative characterization of drought, thereby capturing attenuation and lag effects. Finally, our country-wide drought hazard maps developed from other multivariate indices such as the SPEI (Gupta et al., 2020), as compared to those developed from the univariate SPI (Vittal et al., 2020). This comparison with other papers will be included in the revised manuscript.

Figure S5: Time series of SPI, SSI and MSDI for Marathwada region for 1980 - 2015 (a). Time window for 1980-1984 is expanded in (b). MSDI effectively captures the drought initiation, propagation and termination by correctly characterising drought events whenever either SPI, or SSI, or both fall below a chosen threshold (green horizontal line).

b) L81: "The above two datasets are regridded to a common spatial resolution of 0.5° lat.× 0.5° lon. and rescaled to monthly frequencies for the historical drought hazard assessment." Could you please explain in the supplementary material how this is done?

Re-gridding of the observed datasets to  $0.5^{\circ}$  lat.×  $0.5^{\circ}$  lon resolution is carried out using the Triangulation-based linear interpolation method (Watson and Philip, 1984). This information will be included in the Supplementary material.

Is there an increased spatial variability included by this re-gridding to counteract an averaging effect?

Further, monthly time series of spatial variation in terms of standard deviation of precipitation and soil moisture from their observed and rescaled datasets is now shown in Figure S6 (given below). It is observed that the rescaling of datasets from their parent resolution to  $0.5^{\circ}$  lat.×  $0.5^{\circ}$  lon results in no additional variability.

---

## Author Comment (AC2)

**Responses and Actions taken on Reviewers' Comments**

Journal: Natural Hazards and Earth System Sciences

**Manuscript Reference No.: nhess-2022-18**

**Title**: Evolution of multivariate drought hazard, vulnerability and risk in India under climate change.

**Authors: Venkataswamy Sahana, Arpita Mondal**

We thank the Reviewer for reviewing our manuscript and providing valuable feedback that have helped improve the quality of the work significantly. In this document, we provide a point by point response and actions taken on the comments and suggestions from the reviewers.

**Responses to comments from Referee #2**

**Summary**

1) This paper presents a drought risk assessment for India for a baseline period and for two RCPs and two SSPs. The methodology used seems appropriate for the data used and spatial scale considered. The authors found that drought risk was primarily comprised of the drought vulnerability component, rather than the hazard, these results were shown effectively using bivariate maps. Overall, I found this an interesting paper with results and outcomes that could be useful for drought planning and mitigation at the high level in India. However, I found that the clarity of the paper could be improved and made more expansive making it easier to follow and reproduce elsewhere. Specific examples are discussed below. I recommend that this paper is revised before publication to clarify key methodological points highlighted below.

We thank the reviewer for the positive and encouraging comments. We have addressed the reviewer concerns and provided explanation and clarity on the methods.

**Major comments**

2) I found the description of the methods to calculate the DHI and DVI in the supplementary information unclear, with not enough detail provided on the steps and processes with no further information provided (some examples listed below regarding weighting and standardisation). I would like to see the methods in the main body of the paper expanded. I recommend that all the whole methodology is moved to the main body of the paper, rather than the fundamental steps being in the supplementary information. I would also suggest that Figure 1 in the main

body of the paper is expanded, with more detail and steps added to fully capture the methodological steps described in the paper – this is further discussed below.

We will move the methods on hazard and vulnerability computation to the main manuscript. Further, an example on the drought hazard calculation depicting the weights and ratings for a randomly chosen location is given in Table S2. We have included more details about the methods in Figure 1.

**Drought hazard assessment**

Drought hazard forms an important component of drought risk assessment. Here, we assess the country-wide drought hazard based on the deficiencies in precipitation and soil moisture. Therefore, the multivariate standardized drought index (MSDI) of the non-parametric form is computed using the bivariate case of Gringorten plotting position formula (Gringorten, 1963). The steps involved in the calculation of MSDI is presented below.

The joint probability distribution of the 1-month time scale precipitation (*R*) and soil moisture (*S*) is given by

$$P(R \le r, S \le s) = p \qquad \dots (1)$$

where p represents the joint probability of the precipitation and soil moisture.

2. For the sample size *n*, the count of occurrence of the pair  $(r_i, s_i)$  for  $r_i \le r_k$  and  $s_i \le s_k$  is denoted as  $m_k$ . This count is used to derive the empirical joint probability for the bivariate case with the Gringorten plotting position (Gringorten, 1963) as

$$P(r_k, s_k) = \frac{m_k - 0.44}{n + 0.12} \qquad \dots (2)$$

3. The above empirical joint probability is then standardized to obtain the multivariate index MSDI.

$$MSDI = \varphi^{-1}(P) \qquad \dots (3)$$

where  $\varphi$  is the standard normal distribution function. Since the empirical distributions use ranks of data instead of actual values, the sample size should be sufficiently large.

The method of drought hazard assessment followed in the present study is based on Kim et al. (2015). Hazard is measured as the product of magnitude and the associated frequency of occurrence of an event. The MSDI time series at each region is categorized into four groups

similar to Mckee et al. (1993). These categories are assigned weights according to the magnitude of MSDI value. Higher weights will be assigned to worst (high negative) MSDI values, and vice versa. Further, each weight category is divided into different clusters based on the frequency of occurrence of MSDI values. The total number of clusters for ratings in each MSDI category is determined using the prominent k-means data clustering algorithm. Higher ratings will be assigned to the cluster with high frequency values, and vice versa. The weightage and rating scheme is depicted graphically in Figure 1. In the k-means clustering technique, distance between the data points is computed using the squared Euclidean distance metric. To avoid the convergence to local minima, the k-means algorithm is run with 100 random initial seeds with 10000 iterations. The Calinski-Harabasz Index (CHI) (Caliński and Harabasz, 1974) is used to determine the optimum number of clusters and is given by

$$CHI = \frac{n-K}{K-1} \times \frac{BGSS}{WGSS} \qquad \dots (4)$$

where n= number of data points, K= number of clusters,  $BGSS = \sum_{k=1}^{K} n_k ||G^{\{k\}} - G||^2$  is the between the group scatter,  $G^{\{k\}}$  = centroid of the kth cluster, G = centroid of all the observations,  $WGSS = \sum_{k=1}^{K} WGSS^{\{k\}}$  is within the group scatter and  $WGSS^{\{k\}} = \sum_{i \in I_k} ||M_i^{\{k\}} - G^{\{k\}}||^2$ , where  $M_i^{\{k\}}$  are the observations. The k-means clustering algorithm is driven for 1 to *n* clusters. The number of clusters that gives highest value of CHI is the optimum number of clusters. These optimum number of clusters is used for assigning ratings. The categorized weightages and computed ratings are used to calculate the drought hazard for every region as below.

$$DH = \sum_{i=1}^{t} weights_i \times ratings_i \qquad \dots (5)$$

where t is the length of MSDI time series. Although the weightages and ratings are intrinsically linked, the above scheme assures drought hazard quantification based on magnitudes and frequencies. The *DH* values from Eq 5 are standardized as shown below to obtain *DHI* that varies between 0 and 1.

$$DHI = \frac{DH - DH_{min}}{DH_{max} - DH_{min}} \qquad \dots (6)$$

The weighing and rating scheme to calculate DHI for a randomly chosen grid is given in Table S1. "

| MSDI          | MSDI Class         |   | Frequency of occurence | Rating |
|---------------|--------------------|---|------------------------|--------|
|               | Mild               | 1 | 0.71-0.82              | 6      |
|               |                    |   | 0.60-0.68              | 5      |
| -0.99 to 0.99 |                    |   | 0.49-0.57              | 4      |
|               |                    |   | 0.37-0.46              | 3      |
|               |                    |   | 0.26-0.348             | 2      |
|               |                    |   | 0.18-0.24              | 1      |
|               | Moderate           |   | 0.150-0.15             | 4      |
| -1 to 1.49    |                    | 2 | 0.13-0.13              | 3      |
|               |                    | Z | 0.098-0.098            | 2      |
|               |                    |   | 0.07-0.07              | 1      |
| -1.5 to -1.99 | Severe             | 3 | 0.04-0.04              | 1      |
| -2 or less    | -2 or less Extreme |   | 0.016-0.016            | 1      |

Table S1. Weighting and rating scheme for DHI calculation for a randomly chosen grid (11° lat, 75° lon)

**Drought vulnerability assessment**

Drought vulnerability forms another important component of drought risk assessment. Several aggregation techniques have been employed in the past studies to combine the drought vulnerability indicators to assess drought vulnerability. However, we use the robust method – TOPSIS (Hwang and Yoon, 1981) owing to its lesser rank reversal probabilities (Sahana et al., 2021). The steps involved in drought vulnerability assessment is outlined as below.

1. Standardization of numerical drought vulnerability indicators (irrigation index, water body fraction, groundwater availability, population density and GDP) is carried out such that their values vary between 0 and 1.

$$Std. Indicator = \frac{Indicator - Indicator_{min}}{Indicator_{max} - Indicator_{min}} \qquad \dots (7)$$

Suitable weights are assigned to categorical drought vulnerability indicators (LULC, slope and soil texture), following Thomas et al. (2016) and Sahana et al. (2021) (Table S2). This gives the decision matrix  $n_{ij}$ , where i = 1, 2, ... n represents the number of regions and j = 1, 2, ... m represents the number of drought vulnerability indicators.

2. The above decision matrix  $n_{ij}$  is associated with the indicator weights  $w_j$  obtained from the Analytic Hierarchy Process (AHP) method (Sahana et al., 2021). This gives the weighted decision matrix  $v_{ij}$

$$v_{ij} = w_j n_{ij} \qquad \dots (8)$$

3. Positive  $(A^+)$  and Negative  $(A^-)$  Ideal solution is calculated for each of the indicators.

$$A^{+} = (v_{1}^{+}, v_{2}^{+}, \dots, v_{m}^{+}) = \left[ \left( \max v_{ij} | j \in I \right), \left( \min v_{ij} | j \in J \right) \right] \dots (9)$$

$$A^{-} = (v_{1}^{-}, v_{2}^{-}, \dots, v_{m}^{-}) = \left[ (\min v_{ij} | j \in I), (\max v_{ij} | j \in J) \right] \dots \dots (10)$$

where *I* and *J* are associated with the benefit and cost criteria respectively. Here population density, LULC, slope and soil texture that bear positive correlation with the drought vulnerability are considered as benefit criteria. On the other hand, irrigation index, groundwater availability, waterbody fraction and GDP that bear negative correlation with drought vulnerability are considered as cost criteria.

Positive (d+i) and negative (d-i) separation measures for each region *i* are computed based on A+ and A- (also shown in Figure 1)

$$d_i^+ = \sqrt{\sum_{j=1}^m (\nu_{ij} - \nu_j^+)^2} \qquad \dots (11)$$

$$d_i^- = \sqrt{\sum_{j=1}^m (\nu_{ij} - \nu_j^-)^2} \qquad \dots (12)$$

5. Relative closeness  $(R_i)$  of each region to the Positive Ideal Solution is calculated as

$$R_i = \frac{d_i^-}{d_i^- + d_i^+} \qquad \dots (13)$$

 $R_i$  signifies vulnerability of region *i* to drought. *R* is further standardised to vary between 0 and 1 to obtain drought vulnerability index (DVI)

$$DVI = \frac{R - R_{min}}{R_{max} - R_{min}} \qquad \dots (14)$$

**Table S2. Weightages for categorical vulnerability indicators used for vulnerability assessment (Thomas et al. 2016; Sahana et al. 2021)**

| Vulnerability
indicator | Classification  | Weight | Normalized
Weight |
|----------------------------|-----------------|--------|----------------------|
|                            | Water Body      | 0      | 0                    |
|                            | Barren          | 1      | 0.04                 |
| Landuca                    | Scrub           | 3      | 0.12                 |
| Land use                   | Forest          | 4      | 0.15                 |
|                            | Agriculture     | 8      | 0.31                 |
|                            | Habitation      | 10     | 0.38                 |
|                            | Silty Clay      | 2      | 0.032                |
| Sail                       | Clay            | 3      | 0.048                |
| 5011                       | Silty Clay Loam | 4      | 0.063                |
|                            | Clay Loam       | 5      | 0.079                |

|           | Silt Loam       | 7  | 0.111 |
|-----------|-----------------|----|-------|
|           | Loam            | 9  | 0.143 |
|           | Sandy Clay Loam | 10 | 0.159 |
|           | Sandy Loam      | 11 | 0.175 |
|           | Loamy Sand      | 12 | 0.190 |
|           | 0-1             | 1  | 0.048 |
|           | 1-4             | 2  | 0.095 |
| Slope (%) | 4-6             | 4  | 0.190 |
|           | 6-10            | 6  | 0.286 |
|           | >10             | 8  | 0.381 |

---

## Author Response (AR1)

**Responses and Actions taken on Reviewers' Comments**

**Journal**: Natural Hazards and Earth System Sciences

**Manuscript Reference No**.: nhess-2022- 18

**Title**: Evolution of multivariate drought hazard, vulnerability and risk in India under climate change.

**Authors**: Venkataswamy Sahana, Arpita Mondal

We thank the Reviewers for reviewing our manuscript and providing valuable feedback that have helped improve the quality of the work significantly. In this document, we provide a point by point response and actions taken on the comments and suggestions from the reviewers. (Figure, line, table and page numbers referred to in this document are with respect to the revised manuscript unless mentioned otherwise.)

**Responses to comments from Referee #1**

I have read your manuscript and think it covers an interesting topic. It is clearly the result of a major research effort. Including vulnerability in drought risk analysis is a known challenge, and I agree with you that looking at multiple physical drivers as well as at transient vulnerability are important steps for holistic drought risk assessments. The aim of the study is clearly stated and results are described in detail. However, the research is quite complex and so I think an extra effort is needed to make in understandable for readers of NHESS. I see some conceptual issues, but they may have been caused by a lack of understanding of the method due to its incomplete or undetailed description. In general, I think more of the method could be in the main manuscript and more details to the method (currently lacking) can be described in the supplementary material. Below, I will elaborate on the main points that I think can help improve/clarify the manuscript. In addition, I think the manuscript would benefit from a review by an English language editor, as there are multiple grammar mistakes in the manuscript and I see various possibilities for vocabulary improvements.

We thank the reviewer for the positive and constructive feedback on our work. We have now provided more details and description about the methods and are included in the revised manuscript and the revised supplementary material. Further, we have proof-read the manuscript and corrected for grammar and language wherever necessary. We have addressed the comments provided by you in the below sections.

I haven't listed all grammar / vocabulary mistakes, but here are a few examples from the abstract:

e.g. L7 "a" major threat

e.g. L10 This study investigates and evaluates the change in projected drought risk under future climatic and socio-economic conditions by combining vulnerability and hazard information at a country-wide scale for future climatic and socio-economic conditions

e.g. L18 "are found to be high risk under all scenarios"

Line 7, Page 1

"…pose a major threat…"

Line 10-12, Page 1

"This study investigates and evaluates the change in projected drought risk under future climatic and socio-economic conditions by combining drought hazard and vulnerability projections at a country-wide scale."

Line 19, Page 1

"…are found to be high risk under all scenarios."

e.g. L15-17: Sentence is too long, it is unclear what is meant with "worst-case" scenario

We have now simplified the sentence as

Line 15-18, Page 1

"In the worst-case scenario for drought hazard (RCP2.6-Far future), there is a projected decrease in the area under high or very high drought hazard classes in the country by approximately 7%. Further, the worst-case scenario for drought vulnerability (RCP6.0-SSP2-Near future) shows a 33% rise in the areal extent of high or very high drought vulnerability classes."

I think maybe "The West Utter Pradesh, Haryana, …., regions" are meant rather than "regions of West Utter Pradesh,…"

The sentence is rewritten as

Line 18-19, Page 1

"West Uttar Pradesh, Haryana and West Rajasthan regions are found to be high risk under all scenarios."

In general, in the manuscript there are many sentences that are difficult to understand (too long and/or with too complex structure).

We have simplified complex sentences in the revised manuscript.

Line 432-437, Page 21

"Drought risk projection studies undertaken over the Indian region are based on drought hazard alone, and no consideration has been given to the drought vulnerability component. The present study quantifies the relative contribution of drought hazard and drought vulnerability to the overall drought risk projections under a comprehensive risk framework. Thus, our analysis can aid different stakeholders involved in drought management for adaptation and mitigation plans under changing climate and socio-economic conditions."

Line 299-301, Page 12

"Further, frequency-based soil moisture droughts analysis by Aadhar & Mishra (2020, 2021) and SPEI-based drought frequency analysis by Zhai et al. (2020) show an increased drought frequency in the future period over South Asia compared to the baseline period."

Below, I add some general comments and questions structured following the study aims, highlighting the most pressing questions with respect to the method.

**1. Multivariate drought hazard projection using Multivariate Standardized Drought Index (MSDI) that considers concurrent deficits in precipitation and soil moisture for future warming scenarios.**
a) L81: "*However, droughts can often manifest as a complex interplay of multiple influencing variables necessitating a multivariate approach for characterization of drought hazard (Sahana et al., 2020). For the agrarian country of India, agro-meteorological drought hazard projections should consider deficits in precipitation or soil moisture or both*" I agree looking only at PR is too narrow. It is indeed interesting to look at both, but as far as I understand the method, only events with both a SM-deficit and a PR-deficit are considered. Is this approach justified? I can think of cases where a SM-deficit alone is enough to cause a drought impact – I feel the hazard method does not sufficiently take into account the propagation of drought through the hydrological cycle, which involves attenuation and lag effects. The manuscript

displays different results than other papers: how can it be evidenced that the presented method is better and the results are more reliable than those of other studies?

We would like to clarify that the Multivariate Standardized Drought Index (MSDI) is equally capable of capturing deficits individually in precipitation or soil moisture, or their joint deficit, considering dependence between these two variables. This is a unique advantage of MSDI (Hao and AghaKouchak, 2014) over other univariate indices. This clarification is included in the revised manuscript. Further, we have added Supplementary Figure S6 (given below) that shows how the MSDI is capable of representing the onset, propagation and termination of drought. In this figure, considering -0.8 as the threshold for drought trigger, it seen that whenever either the SPI or the SSI falls below this threshold, MSDI covers the critical trajectory and offers a conservative characterization of drought, thereby capturing attenuation and lag effects. Finally, our country-wide drought hazard map for the baseline period from the present study (Figure 3) matches well with hazard maps developed from other multivariate indices such as the SPEI (Gupta et al., 2020), as compared to those developed from the univariate SPI (Vittal et al., 2020). This comparison with other papers is now included in the revised manuscript.

[Figure]

Figure S6. Time series of SPI, SSI and MSDI for Marathwada region for 1980 – 2015 (a). Time window for 1980-1984 is expanded in (b). MSDI effectively captures the drought initiation, propagation and termination by correctly characterising drought events whenever either SPI, or SSI, or both fall below a chosen threshold (green horizontal line).

Line 158-161, Page 6

"Further, MSDI is capable of capturing deficits individually in precipitation or soil moisture, or their joint deficit, considering dependence between these two variables. This is a unique advantage of MSDI (Hao and AghaKouchak, 2014) over other univariate indices. In the Figure S6, considering -0.8 as the threshold for drought trigger, it seen that whenever either the SPI or the SSI falls below this threshold, MSDI covers the critical trajectory and offers a conservative characterization of drought, thereby capturing attenuation and lag effects."

Line 270-272, Page 11

"The MSDI-based drought hazard maps developed for the baseline period matches well with hazard maps developed from other multivariate indices such as SPEI (Gupta et al., 2020), as compared to those developed from the univariate SPI (Vittal et al., 2020)."

b) L81: *"The above two datasets are regridded to a common spatial resolution of 0.5° lat.× 0.5° lon. and rescaled to monthly frequencies for the historical drought hazard assessment."* Could you please explain in the supplementary material how this is done?

Re-gridding of the observed datasets to 0.5° lat.× 0.5° lon resolution is carried out using the Triangulation-based linear interpolation method (Watson and Philip, 1984). This information is included in the revised manuscript.

Line 83-85, Page 3

"Re-gridding of the observed datasets to 0.5° lat.× 0.5° lon resolution is carried out using the Triangulation-based linear interpolation method (Watson and Philip, 1984)."

Is there an increased spatial variability included by this re-gridding to counteract an averaging effect?

Further, monthly time series of spatial variation in terms of standard deviation of precipitation and soil moisture from their observed and rescaled datasets is now shown in Figure S1 (given below). It is observed that the rescaling of datasets from their parent resolution to 0.5° lat.× 0.5° lon results in no additional variability.

Line 85-87, Page 3

"Further, monthly time series of spatial variation in terms of standard deviation of precipitation and soil moisture from their observed and rescaled datasets is shown in Figure S1. It is observed

that the rescaling of datasets from their parent resolution to 0.5° lat.× 0.5° lon results in no additional variability."

[Figure]

Figure S1. Standard deviation of the country-wide cumulated observed and rescaled datasets of precipitation (top panel) and soil moisture (bottom panel).

c) L75 + 83: "T*he drought hazard assessment for baseline period (1980-2015) requires observed hydroclimatic variables*" + "*In order to evaluate the projected drought hazard over India, the projected precipitation and soil moisture data at a spatial resolution of 0.5° lat.× 0.5° lon. is obtained from the Inter-Sectoral Impact Model Intercomparison Project (ISIMIP) (Warszawski et al., 2014). The historical (1980-2005) and projected (2006-2099) data from available GCMs namely ....*" How did you deal with the overlapping time period between observed and modelling data? Was, for example, the Delta Method (projecting the difference between modelled historic and projected onto the observed data; or; projecting to difference between observed and modelled historic onto the projected data) applied? I do not find information on how the final hazard dataset is constructed – so I suggest adding this to the supplementary material.

Here, historical (1980-2015) hazard maps are generated only from observed datasets - IMD for precipitation and MERRA for soil moisture. The projected Near (2021-2060) and Far (2061-2099) future hazard is obtained from the GCMs. We do not use the Delta Method or any such procedure to compare and 'correct' data from the GCMs, since the ISIMIP uses precipitation

data that has already been downscaled and bias-corrected with respect to global level observed precipitation from EartH2Observe observations, WFDEI and ERA-Interim data. Further, for obtaining projections of soil moisture, ISIMIP employs the global vegetation model, LPJmL, that is capable of representing fine resolution physical processes using carbon, water and energy balance equations (Schaphoff et al., 2018) under a changed climate, thereby, offering a significant improvement over simplistic data-based approaches such as the Delta Method. This information is included in the revised manuscript under the Section 2.1.1.

Line 92-94, Page 3-4

"The daily precipitation data (kg m-2 s-1) is already been downscaled and bias corrected with respect to global level observed precipitation from EartH2Observe observations, WFDEI and ERA-Interim data Merged and Bias-corrected for ISIMIP (EWEMBI)."

Line 97-99, Page 4

"As a part of ISIMIP2b experiments, the LPJmL impact model (Sitch et al., 2003), a global vegetation model that is capable of representing fine resolution physical processes using carbon, water and energy balance equations (Schaphoff et al., 2018) under a changed climate, is driven by the bias-corrected GCM precipitation to simulate the root-zone soil moisture (kg m-2)."

"Figure 1. Framework to assess drought risk evolution. Monthly rainfall and monthly soil moisture is used to compute multivariate standardized drought index (MSDI). Weights and ratings system of MSDI is adopted to further compute drought hazard index (DHI). Multi-criteria decision making technique – TOPSIS is used to calculate drought vulnerability index (DVI) considering eight drought vulnerability indicators. The product of DHI and DVI is the drought risk index (DRI). Drought risk assessment is carried out for Baseline period (1980-2015), Near future (2021-2050) and Far future (2061-2100) for various climate and socio-economic scenarios."

It would be nice to show with some figures how the ISIMIP data and the used observed IMD Pr and MERRA SM data compare?

Further, based on the reviewer's suggestion, we carry out an additional analysis for evaluation of ISIMIP simulations with respect to observed precipitation and soil moisture data, and present

the results of such evaluation in Figure S4 in the revised manuscript (given below). The performance of all the ISIMIP models are comparable with that of the observed data, except for the soil moisture during monsoon months. The lowered soil moisture estimates from LPJmL model (ISIMIP experiments) simulations compared to the MERRA-Land soil moisture observations for the monsoon months could be due to overestimation of LPJmL's simulated runoff (Zaherpour et al., 2018).

[Figure]

[Figure]

Figure S4. Observed and ISIMIP-model simulated climatology of country-wide average monthly precipitation and soil moisture for the period 1980-2005.

Line 101-106, Page 4

"The observed and simulated country-wide average of monthly precipitation and soil moisture for the period 1980-2005 is presented in Figure S4. The performance of all the ISIMIP models are comparable with that of the observed data, except for the soil moisture during monsoon months. The lowered soil moisture estimates from LPJmL model (ISIMIP experiments) simulations compared to the MERRA-Land soil moisture observations for the monsoon months could be due to overestimation of LPJmL's simulated runoff (Zaherpour et al., 2018)."

d) L93 "*The spatial pattern of annual mean surface soil moisture for India from the LPJmL impact model is consistent with the satellite-based Essential Climate Variable soil moisture product (Gu et al., 2019)*." Is this ECV similar to the MERRA Land data used? Or how is it connected to the data used?

We have performed an additional analysis to compare the LPJmL soil moisture dataset with MERRA. Therefore, we remove the statement regarding comparison of LPJmL soil moisture dataset with respect to ECV. We refer to our Figure S4 and included the following discussion in the revised manuscript.

Line 103-106, Page 4

"The performance of all the ISIMIP models are comparable with that of the observed data, except for the soil moisture during monsoon months. The lowered soil moisture estimates from LPJmL model (ISIMIP experiments) simulations compared to the MERRA-Land soil moisture observations for the monsoon months could be due to overestimation of LPJmL's simulated runoff (Zaherpour et al., 2018)."

e) L95 "*Although the simulated soil moisture data underestimates the monsoon months' soil moisture (June, Jul, Aug, Sep) during the historic period (1980-2005), we did not perform the bias correction, since we intend to capture the variability in the soil moisture rather than their magnitudes for drought index calculation*" – can you please add graphs / maps to show this in the supplementary please?

We have included Figure S4 (given in Comment #1c) in the revised Supplementary material.

f) In the Supplementary Material (drought hazard assessment and S1): Is the co-occurrence – covariance of Pr and SM modelled per ensemble member after which the mean of the DH value is calculated? Or are ensemble mean / median PR and SM used to calculate the DH value?

The ensemble mean of monthly precipitation and soil moisture from different GCMs is computed. Further, these ensemble mean monthly precipitation and soil moisture time series is used to calculate the MSDI and DHI values.

Line 111-112, Page 4

"The ensemble mean of monthly precipitation and soil moisture from different GCMs is computed. Further, these ensemble mean monthly precipitation and soil moisture time series is used for drought hazard assessment."

What are rk and sk in formula 2 – are they thresholds for droughts in SM and PR?

$r_k$ and $s_k$ denote the actual precipitation and soil moisture values for $k^{th}$ observation. In Gringorten plotting position method, the number of occurrences ($m_k$) of precipitation and soil

moisture pair below $r_k$ and $s_k$ from the whole set of observations is used to calculate empirical joint probability for k$^{th}$ observation.

Line 165-168, Page 6

"For the sample size $n$, the count of occurrence of the pair $(r_i, s_i)$ for $r_i \leq r_k$ and $s_i \leq s_k$ is denoted as $m_k$. Here $r_k$ and $s_k$ denote the actual precipitation and soil moisture values for kth observation. The number of occurrences $(m_k)$ of precipitation and soil moisture pair below $r_k$ and $s_k$ from the whole set of observations is used to calculate empirical joint probability for k$^{th}$ observation based on bivariate Gringorten plotting position (Gringorten, 1963) as"

g) In the Supplementary Material (drought hazard assessment): Until "*The MSDI series at each region is categorized into four groups similar to Mckee et al. (1993)."* I could follow the description, then it becomes unclear à please add more detail (e.g., on the weighing and rating: I do not understand why nor how this is done) and please add some examples to showcase and justify the method.

We rewrote the methodology for drought hazard assessment and is now included in the main manuscript. An example on the calculation of drought hazard is also added.

Line 174-197, Page 6-7

"The method for drought hazard assessment followed in the present study is based on Kim et al. (2015). Hazard is measured as the product of magnitude and the associated frequency of occurrence of an event. The MSDI time series at each region is categorized into four groups similar to Mckee et al. (1993). These categories are assigned weights according to the magnitude of MSDI value. Higher weights will be assigned to worst (high negative) MSDI values, and vice versa. Further, each weight category is divided into different clusters based on the frequency of occurrence of MSDI values. The total number of clusters for ratings in each MSDI category is determined using the prominent k-means data clustering algorithm. Higher ratings will be assigned to the cluster with high frequency values, and vice versa. The weightage and rating scheme is depicted graphically in Figure 1. In the k-means clustering technique, distance between the data points is computed using the squared Euclidean distance metric. To avoid the convergence to local minima, the k-means algorithm is run with 100 random initial seeds with 10000 iterations. The Calinski-Harabasz Index (CHI) (Caliński and Harabasz, 1974) is used to determine the optimum number of clusters and is given by

$$CHI = \frac{n-K}{K-1} \times \frac{BGSS}{WGSS} \qquad \ldots (4)$$

where $n=$ number of data points, $K=$ number of clusters, $BGSS = \sum_{k=1}^{K} n_k ||G^{\{k\}} - G||^2$ is the between the group scatter, $G^{\{k\}}$ = centroid of the k$^{\text{th}}$ cluster, $G$ = centroid of all the observations, $WGSS = \sum_{k=1}^{K} WGSS^{\{k\}}$ is within the group scatter and $WGSS^{\{k\}} = \sum_{i \in I_k} ||M_i^{\{k\}} - G^{\{k\}}||^2$, where $M_i^{\{k\}}$ are the observations. The k-means clustering algorithm is driven for 1 to $n$ clusters. The number of clusters that gives highest value of CHI is the optimum number of clusters. These optimum number of clusters is used for assigning ratings. The categorized weightages and computed ratings are used to calculate the drought hazard for every region as below.

$$DH = \sum_{i=1}^{t} weights_i \times ratings_i \qquad \ldots (5)$$

where $t$ is the length of MSDI time series. Although the weightages and ratings are intrinsically linked, the above scheme assures drought hazard quantification based on magnitudes and frequencies. The $DH$ values from Eq 5 are standardized as shown below to obtain $DHI$ that varies between 0 and 1.

$$DHI = \frac{DH - DH_{min}}{DH_{max} - DH_{min}} \qquad \ldots (6)$$

The weighing and rating scheme to calculate DHI for a randomly chosen grid is given in Table S2. "

Table S2. Weighting and rating scheme for DHI calculation for a randomly chosen grid (11˚ lat, 75˚ lon).

| MSDI | Class | Weight | Frequency of occurence | Rating |
|---|---|---|---|---|
| -0.99 to 0.99 | Mild | 1 | 0.71-0.82 | 6 |
| | | | 0.60-0.68 | 5 |
| | | | 0.49-0.57 | 4 |
| | | | 0.37-0.46 | 3 |
| | | | 0.26-0.348 | 2 |
| | | | 0.18-0.24 | 1 |
| -1 to -1.49 | Moderate | 2 | 0.150-0.15 | 4 |
| | | | 0.13-0.13 | 3 |
| | | | 0.098-0.098 | 2 |
| | | | 0.07-0.07 | 1 |

| -1.5 to -1.99 | Severe | 3 | 0.04-0.04 | 1 |
| -2 or less | Extreme | 4 | 0.016-0.016 | 1 |

h) Supplementary "*Further, each category is organised into sub-groups based on the occurrence probabilities of the selected category. While the weightages are assigned to MSDI categories to account for drought magnitude, ratings are assigned to the sub-groups of each MSDI category to account for drought occurrence probability.*" -> which categories? And how does dividing based on occurrence probabilities differ from McKee et al? That is what they do, too, no? I do not understand why both are needed since they (intensity and probability) are intrinsically linked.

The division of MSDI series into different drought groups based on Mckee et al. (1993) gives the magnitude of drought events alone. However, hazard is a measure of magnitude of the event as well as its associated frequency. Therefore from the available MSDI series, it is required to discretize magnitude (weights) and occurrence probability (ratings) (Kim et al., 2015), though they are intrinsically linked. This is clarified in the methods section.

Line 193-194, Page 7

"Although the weightages and ratings are intrinsically linked, the above scheme assures drought hazard quantification based on magnitudes and frequencies."

**2. Drought vulnerability projection considering combinations of RCP and SSP scenarios, using a list of drought vulnerability indicators that represent exposure, sensitivity and adaptive capacity components.**

a) The manuscripts' understanding of vulnerability (including exposure) does not fully match the understanding of this concept by the sources cited (IPCC AR5) and does not match L36 (although it is true other authors see exposure as part of vulnerability – so I suggest look up other scholars who also include exposure as part of the vulnerability quantification). Besides, with respect to the chosen vulnerability factors, I think multiple interesting other social, economic vulnerability indicators could have been selected (e.g., Meza et al 2020 https://nhess.copernicus.org/articles/20/695/2020/ )

We agree with the reviewer that our characterization of drought vulnerability is not fully consistent with the IPCC (AR5)'s recommended definition of drought risk. 
[revised manuscript text omitted]

b) Table 1: I do not always follow the reasoning regarding the relevance (I would not say population density and land use cover are proxies for social vulnerability) but more importantly: I would like to see some more information about how these indices are calculated (population density is pop sum / area; but how is the water bodies fraction calculated, or the irrigation index? How does the water holding capacity positively influence the vulnerability?).

The drought vulnerability indicators chosen in this study, their sources, spatial and temporal distribution, units, method for data generation, relevance to the drought vulnerability, correlation with drought vulnerability, and previous studies who have employed such data for regional/national/global drought vulnerability studies are presented in Table S1 from Sahana et al. (2021). The population density is given as the population count by area, with its units as persons/km$^2$. Irrigation index is the ratio of total irrigated area to the total cropped area. Further, soil textural properties range from clayey to loamy, with clayey soils having higher water holding capacity compared to loamy soils. Hence loamy soils are more vulnerable to drought compared to clayey soils. Also, weightages for different categories of soil texture is presented in Table S1. All the above information is now updated in Table 1.

Line 126-127, Page 5

"Further, the weightages for the categorical vulnerability indicators for drought vulnerability assessment is adopted from (Ekrami et al., 2016; Sahana et al., 2021; Thomas et al., 2016), and is given in Table S1."

Line 116-118, Page 4

"The country-wide drought vulnerability indicators adopted for drought vulnerability assessment are listed in Table 1, along with their sources, sources, spatial and temporal distribution, units, method of data generation, relevance and correlation to drought vulnerability for both the observed (around the year 2010) and projected datasets (2005-2100)."

Besides, I see different sources used to the observed versus projected situation: how is consistency ensured?

The following discussion regarding the consistency of the datasets between baseline and projected datasets is now included in the revised manuscript.

Line 135-149, Page 5

"Drought vulnerability indicators such as population density and GDP for the year 2010 from SSP2 pathway are comparable with their respective observed dataset, with small/negligible difference between the observed and SSP-simulated datasets (Figure S5). Further, drought vulnerability indicators such as groundwater availability, irrigation index and waterbody fraction for the projected period are not directly available. Hence, these indicators are proxied by their representative indicators (Table 1) using multiple linear regression (MLR). Consequently, irrigation ratio, groundwater availability and water body fraction for the projected period are derived based on relationships between them and the representative variables in the baseline period, and therefore consistency is ensured. The Land Use Harmonization (LUH) (Chini et al., 2014) dataset provides the fractional land use classes for the time period 1500-2100. The historical maps of crop and pasture data from HYDE 3.1 (Hurtt et al., 2011), and estimates of historical national wood harvest and of shifting cultivation are used as input for 1500-2005. Further, the projections of LULC for 2005-2100 are based on the Integrated Assessment Model (IAM) implementations of the RCPs. Each IAM for different RCPs are used as input to the Earth System Models (ESMs) for future carbon-climate projections. Therefore, LULC scenarios are based on RCPs. LUH is a credible dataset for LULC projection and has been previously used for drought risk projection in South-Asian region (Chou et al., 2019). Further, LULC projections can also be derived based on the land use models, using past LULC data and socio-economic factors driving the land use change. However, development of such models at country scale is beyond the scope of the present study."

[Figure]

Figure S5. Difference between the observed and SSP2 pathway dataset at the year 2010 for a) Population density and b) GDP.

c) In the supplementary material: Please repeat the weights of Thomas and Sahana for the vulnerability indicators

Done.

Table S1. Weightages for categorical vulnerability indicators used for vulnerability assessment (Thomas et al. 2016; Sahana et al. 2021)

| Vulnerability indicator | Classification | Weight | Normalized Weight |
|---|---|---|---|
| Land use | Water Body | 0 | 0 |
| | Barren | 1 | 0.04 |
| | Scrub | 3 | 0.12 |
| | Forest | 4 | 0.15 |
| | Agriculture | 8 | 0.31 |
| | Habitation | 10 | 0.38 |
| Soil | Silty Clay | 2 | 0.032 |
| | Clay | 3 | 0.048 |
| | Silty Clay Loam | 4 | 0.063 |
| | Clay Loam | 5 | 0.079 |
| | Silt Loam | 7 | 0.111 |
| | Loam | 9 | 0.143 |
| | Sandy Clay Loam | 10 | 0.159 |
| | Sandy Loam | 11 | 0.175 |
| | Loamy Sand | 12 | 0.190 |

| | | | |
|---|---|---|---|
| | 0-1 | 1 | 0.048 |
| | 1-4 | 2 | 0.095 |
| **Slope (%)** | 4-6 | 4 | 0.190 |
| | 6-10 | 6 | 0.286 |
| | >10 | 8 | 0.381 |

**3. Drought risk projection integrating hazard and drought vulnerability information.**

a) In general, there is no validation of the presented risk approach since the past risk analysis (1980-2015) is not compared with observed risk / reported impacts. This should be done in order to give credibility to the method, or – if impossible – be addressed in the discussion section.

Validation of the drought risk map for the baseline period (1980-2015) has been carried out by Sahana et al. (2021) (see Suppl. Figure S3 of Sahana et al., 2021), based on the disaster data in terms of number of people affected. It is noted that parts of Rajasthan, Madhya Pradesh, Maharashtra, Orissa and Tamil Nadu, Kerala, Chattisgarh, Haryana, Himachal Pradesh, Chandigarh, Assam and Nagaland that are under moderate to severe drought risk category, have experienced moderate to worst drought disaster. The above information on validation of drought risk is included in the revised manuscript.

Line 367-371, Page 17

"It is to be noted that the validation of drought risk map for the baseline period (1980-2015) has been carried out by Sahana et al. (2021), based on the based on the disaster data in terms of number of people affected. It is noted that parts of Rajasthan, Madhya Pradesh, Maharashtra, Orissa and Tamil Nadu, Kerala, Chattisgarh, Haryana, Himachal Pradesh, Chandigarh, Assam and Nagaland that are under moderate to severe drought risk category, have experienced moderate to worst drought disaster."

b) L119 "*Drought risk values computed using Eq 1 are further standardized to obtain the Drought Risk Index (DRI)."* Can you please elaborate how this is done? Two standardized indices are multiplied so I do not see the need to standardize the result again – this introduces some loss of information?

The equation and need for standardization is now explained in the revised manuscript.

Line 237-240, Page 8

"Standardization of drought risk at each grid is carried out using the equation

$$DRI = \frac{Rsik - Risk_{min}}{Risk_{max} - Risk_{min}} \qquad \ldots (16)$$

Standardization is performed such that the values are distributed between 0 and 1, so as to classify different risk categories."

c) The effect of climate change is taken into account in two ways: by changing vulnerability (multiple vulnerability indicators are based on average water availability) and by changing occurrence. I think this is interesting but it is a pity that social vulnerability factors, influenced by socio-economic development, are not taken into account – this might have changed the vulnerability trend hence risk trend. Would it be possible to account for this?

The population density and GDP indicators, considered in the present study, accounts for social vulnerability, and the change in these indicators are accounted for vulnerability projections. Therefore, we do not agree that our study does not take into account social vulnerability factors. However, we do agree with the reviewer that the study would be comprehensive with the inclusion of other socio-economic indicators. However, for a densely-populated and rapidly-developing nation such as India, acquisition of reliable datasets on these indicators is often challenging. Most importantly, unavailability of projections of these indicators over the Indian region limits their use in this study, since our primary goal is to compare baseline drought risk with that under future projected climate change. This discussion is included in the revised manuscript.

Line 322, Page 15

[revised manuscript text omitted]

e) Fig2: I do not understand why land cover changes based on RCPs? Shouldn't this be SSP? Besides, I wonder why baseline (1980-2015) isn't shown? Now it is indicated as "2010" but that seems inconsistent with the method section.

The projections of LULC for 2005-2100 are based on the Integrated Assessment Model (IAM) implementations of the RCPs, and not based on SSPs (Chini et al., 2014). Further, in the revised manuscript, we have included a discussion on why baseline LULC time series is not shown in

Figure 2. For correct representation of vulnerability indicators, we will replace their time series with bar graphs for the years 2010, 2060 and 2099 (Figure 2b, see comment #3d).

Line 143-146, Page 5

"Further, the projections of LULC for 2005-2100 are based on the Integrated Assessment Model (IAM) implementations of the RCPs. Each IAM for different RCPs are used as input to the Earth System Models (ESMs) for future carbon-climate projections. Therefore, LULC scenarios are based on RCPs."

Line 129-134, Page 5

"In general, socio-economic development is a slow process, and takes time to reflect in terms of significant changes in the socio-economic indicators (Dellink et al., 2017). Further, majority of the drought vulnerability/risk studies across the globe have adopted static vulnerability assessment that represent drought vulnerability snapshot in time (Hagenlocher et al., 2019). Therefore, we used the static vulnerability indicators for the year 2010, 2060 and 2099 to quantify drought vulnerability for the baseline, Far future and Near future period respectively."

**4. Development of bivariate choropleth plots under future scenarios to quantify the individual roles of climate and societal changes in driving drought risk**

a) This is a good way of visualising the results; but I would suggest to change the colour classes since now on e.g., the RCP6.0 near future, barely any variance is visible.

The RCP6.0 Near future results in mostly high vulnerability and low hazard regions. It is to be noted that the low variability in this scenario is due to data and not necessarily due to selected color scheme, since parts of Bihar and Telangana are distinctive with moderate vulnerability and moderate hazard. For a better comparison of the future scenarios with the baseline period, we would retain the existing bivariate color scheme.

**5. Identification of regions and zones that are expected to be under worst drought risk conditions in the near and far future**

a) (Make sure that in the discussion, the results are compared with papers who have a similar conception of vulnerability – or discuss the difference – because that might also be the cause of the diverging results)

We have now included the following discussion in Section 3.3.1 of the revised manuscript.

Line 371-373, Page 17

"Further, the drought risk estimates for the baseline period from the present study compares well with regional-scale drought risk studies in India such as those for Andhra Pradesh (Murthy et al., 2015), Bearma basin (Thomas et al., 2016), Maharashtra (Swami and Parthasarathy, 2021)."

Line 389-392, Page 18

"It is to be noted that the water availability projections for India by Koutroulis et al. (2019) show decreasing drought risk with time, as opposed to the increasing drought risk from the present study. The choice of climate change scenarios and climate models by Koutroulis et al. (2019) could be a possible reason for such difference."

**Responses to comments from Referee #2**

**Summary**

1) This paper presents a drought risk assessment for India for a baseline period and for two RCPs and two SSPs. The methodology used seems appropriate for the data used and spatial scale considered. The authors found that drought risk was primarily comprised of the drought vulnerability component, rather than the hazard, these results were shown effectively using bivariate maps. Overall, I found this an interesting paper with results and outcomes that could be useful for drought planning and mitigation at the high level in India. However, I found that the clarity of the paper could be improved and made more expansive making it easier to follow and reproduce elsewhere. Specific examples are discussed below. I recommend that this paper is revised before publication to clarify key methodological points highlighted below.

We thank the reviewer for the positive and encouraging comments. We have addressed the reviewer concerns and provided explanation and clarity on the methods.

**Major comments**

2) I found the description of the methods to calculate the DHI and DVI in the supplementary information unclear, with not enough detail provided on the steps and processes with no further information provided (some examples listed below regarding weighting and standardisation). I would like to see the methods in the main body of the paper expanded. I recommend that all the whole methodology is moved to the main body of the paper, rather than the fundamental steps being in the supplementary information. I would also suggest that Figure 1 in the main body of the paper is expanded, with more detail and steps added to fully capture the methodological steps described in the paper – this is further discussed below.

We have now moved methods on hazard and vulnerability computation to the main manuscript. Further, an example on the drought hazard calculation depicting the weights and ratings for a randomly chosen location is given in Table S2. We have included more details about the methods in Figure 1.

Line 152-197, Page 5-7

[revised manuscript text omitted]

Line 198-228, Page 7-8

**"2.2.2 Drought vulnerability assessment**

Drought vulnerability forms another important component of drought risk assessment. Several aggregation techniques have been employed in the past studies to combine the drought vulnerability indicators to assess drought vulnerability. However, we use the robust method – TOPSIS (Hwang and Yoon, 1981) owing to its lesser rank reversal probabilities (Sahana et al., 2021). The steps involved in drought vulnerability assessment is outlined below.

1. Standardization of numerical drought vulnerability indicators (irrigation index, water body fraction, groundwater availability, population density and GDP) is carried out such that their values vary between 0 and 1.

$$Std. Indicator = \frac{Indicator - Indicator_{min}}{Indicator_{max} - Indicator_{min}} \qquad ... (7)$$

Suitable weights are assigned to categorical drought vulnerability indicators (LULC, slope and soil texture), following Thomas et al. (2016) and Sahana et al. (2021) (Table S1). This gives the decision matrix $n_{ij}$, where $i = 1,2, ... n$ represents the number of regions and $j = 1,2, ... m$ represents the number of drought vulnerability indicators.

2. The above decision matrix $n_{ij}$ is associated with the indicator weights $w_j$ obtained from the Analytic Hierarchy Process (AHP) method (Sahana et al., 2021). This gives the weighted decision matrix $v_{ij}$

$$v_{ij} = w_j n_{ij} \qquad ... (8)$$

3. Positive ($A^+$) and Negative ($A^-$) Ideal solution is calculated for each of the indicators.

$$A^+ = (v_1^+, v_2^+, \dots v_m^+) = \left[(\max v_{ij}|j \in I), (\min v_{ij}|j \in J)\right] \qquad \dots (9)$$

$$A^- = (v_1^-, v_2^-, \dots v_m^-) = \left[(\min v_{ij}|j \in I), (\max v_{ij}|j \in J)\right] \qquad \dots (10)$$

where $I$ and $J$ are associated with the benefit and cost criteria respectively. Here population density, LULC, slope and soil texture that bear positive correlation with the drought vulnerability are considered as benefit criteria. On the other hand, irrigation index, groundwater availability, waterbody fraction and GDP that bear negative correlation with drought vulnerability are considered as cost criteria.

4. Positive ($d_i^+$) and negative ($d_i^-$) separation measures for each region $i$ are computed based on $A^+$ and $A^-$ (also shown in Figure 1)

$$d_i^+ = \sqrt{\sum_{j=1}^{m} (v_{ij} - v_j^+)^2} \qquad \dots (11)$$

$$d_i^- = \sqrt{\sum_{j=1}^{m} (v_{ij} - v_j^-)^2} \qquad \dots (12)$$

5. Relative closeness ($R_i$) of each region to the Positive Ideal Solution is calculated as

$$R_i = \frac{d_i^-}{d_i^- + d_i^+} \qquad \dots (13)$$

$R_i$ signifies vulnerability of region $i$ to drought. $R$ is further standardised to vary between 0 and 1 to obtain drought vulnerability index (DVI)

$$DVI = \frac{R - R_{min}}{R_{max} - R_{min}} \qquad \dots (14)"$$

Table S1. Weightages for categorical vulnerability indicators used for vulnerability assessment (Thomas et al. 2016; Sahana et al. 2021)

| Vulnerability indicator | Classification | Weight | Normalized Weight |
|---|---|---|---|
| Land use | Water Body | 0 | 0 |
| | Barren | 1 | 0.04 |
| | Scrub | 3 | 0.12 |
| | Forest | 4 | 0.15 |
| | Agriculture | 8 | 0.31 |
| | Habitation | 10 | 0.38 |
| Soil | Silty Clay | 2 | 0.032 |
| | Clay | 3 | 0.048 |
| | Silty Clay Loam | 4 | 0.063 |
| | Clay Loam | 5 | 0.079 |
| | Silt Loam | 7 | 0.111 |
| | Loam | 9 | 0.143 |
| | Sandy Clay Loam | 10 | 0.159 |
| | Sandy Loam | 11 | 0.175 |
| | Loamy Sand | 12 | 0.190 |
| Slope (%) | 0-1 | 1 | 0.048 |
| | 1-4 | 2 | 0.095 |
| | 4-6 | 4 | 0.190 |
| | 6-10 | 6 | 0.286 |
| | >10 | 8 | 0.381 |

[Figure]

Figure 1. Framework to assess drought risk evolution. Monthly rainfall and monthly soil moisture is used to compute multivariate standardized drought index (MSDI). Weights and ratings system of MSDI is adopted to further compute drought hazard index (DHI). Multi-criteria decision making technique – TOPSIS is used to calculate drought vulnerability index (DVI) considering eight drought vulnerability indicators. The product of DHI and DVI is the drought risk index (DRI). Drought risk assessment is carried out for Baseline period (1980-2015), Near future (2021 2050) and Far future (2061-2100) for various climate and socio-economic scenarios.

3) Use of terminology – literature usually talks about vulnerability factors – i.e. the factors that make a person or location vulnerable to drought impacts. Also regarding vulnerability terminology, you mention components of vulnerability in the introduction (L35-38), how do the indicators (or factors) you used map onto these? This could be included in Table 1. Did you consider using for example, WorldPop data such that the vulnerability assessment could be disaggregated by sex?

Table 1 is revised to represent sensitivity, adaptive capacity and exposure components of the drought vulnerability indicators in terms of their socio-economic, physical and infrastructural aspects. We agree with the reviewer that the study would be comprehensive with the inclusion of other socio-economic indicators. However, for a densely-populated and rapidly-developing nation such as India, acquisition of reliable datasets on these indicators is often challenging. Most importantly, unavailability of projections of these indicators over the Indian region limits their use in this study, since our primary goal is to compare baseline drought risk with that under future projected climate change. This discussion is included in the revised manuscript.

Line 122-126, Page 4-5

[revised manuscript text omitted]

A final question on the factors used, how do the vulnerability factors selected address the exposure component? Although a population may be vulnerable to the impact of drought, they may not be exposed to drought or may be exposed to a lower severity of drought hazard than in other locations, for example.

In this study, population density, land use and soil type constitute the exposure indicators for drought vulnerability assessment (Table 1). We agree with the reviewer that a highly vulnerable population might be exposed to mild droughts or no droughts at all. These differences are precisely accounted in the overall drought risk estimates, since we characterize risk as a combination of vulnerability and hazard. For eg., if the hazard is low in a region, it is likely to be classified as 'low to moderate' in terms of drought risk, despite having high vulnerability. This discussion is included in the revised manuscript.

Line 241-243, Page 8

"Further, circumstances such as highly vulnerable population being exposed to mild droughts or no droughts at all may arrive, and are handled well due to the integrated assessment of drought risk. For eg., if the hazard is low in a region, it is likely to be classified as 'low to moderate' in terms of drought risk, despite having high vulnerability."

4) The baseline period used (1980-2015) excludes the past six years, excluding significant drought events in 2016-2018 and 2021. Is there a way that updated precipitation data could be obtained and analysed to include these recent events?

We have used hydro-climatic and socio-economic variables for the period 1980-2015 to ensure overlap between drought hazard and vulnerability indicators for the baseline period. The overall risk estimates may be misleading if the exposure, adaptive capacity and sensitivity indicators are overlayed on hazard events of dissimilar timelines. Further, our analysis takes into account the major drought episodes of 1982, 1984, 1986, 1987, 1989, 1991, 2000, 2002 and 2015 as identified by earlier study (Sahana et al., 2020). Among them, the major drought episodes of 2002 and 2015 had severely impacted multiple sectors of the country. For example, agricultural contributions to GDP dipped by 3.1% along with heavy agricultural income losses in the 2002 drought event

(DownToEarth, 2015), while the 2015 drought event affected over 330 million people (BBC India, 2016), and reservoir levels dropped to minimal values (The Times of India, 2016). These events contribute to the hazard assessment in this study. Additional inclusion of subsequent droughts of 2016-2018 and 2020 is unlikely to significantly alter our conclusions.

5) L140-142: the first sentence here states that variability of the two scenarios increased over time, but the second states that the baseline period is more variable than the projected period. I am not sure how both of these statements can be true, nor am I convinced I can see these difference in the time series for precipitation or soil moisture. Please clarify this point further.

Figure 2 is updated to include the time series of precipitation and soil moisture for the baseline period as well (1980-2015). It is observed that the variability of both the variables in the projected period increases with time. However, it is evident from Figure 2, that the variability in the hydro-climatic variables in the baseline period is high compared to the projected period. This is clarified in the revised manuscript.

Line 280-285, Page 11

"From Figure 2a, S1 and S2, we see that precipitation and soil moisture for the projected period show an increasing trend. Further, it is to be noted that the hazard assessment using MSDI is based on the long term mean and variability of these drought indicators under a probabilistic analysis framework, and not necessarily the magnitudes of precipitation and soil moisture. Here we see that the projections of these indicators exhibit lower variability compared to the baseline period (Figure 2a). Therefore, it is observed that many regions undergo transition from high hazard to low hazard."

[Figure]

Figure 2. Datasets used for drought risk assessment. a) Projected hydro-climatic variables such as monthly precipitation and monthly soil moisture are used for drought hazard assessment. b) Projected drought vulnerability indicators such as irrigation index, water body fraction, groundwater availability, population, GDP and land use land cover, along with static drought vulnerability indicators such as slope and soil texture are used for drought vulnerability assessment. Datasets for projected period are divided into Near future (2021-2060) and Far future (2061-2100) to check the evolution of drought risk.

This also applies to Table 3 and the text in L199-205, and Table 4 and text L232-233.

Table 3 and Table 4 represent the transition of drought vulnerability/risk from one class of vulnerability/risk from baseline to another class of vulnerability/risk in the future. In general,

socio-economic development is a slow process, and takes time to reflect in terms of significant changes in the socio-economic indicators (Dellink et al., 2017). Further, majority of the drought vulnerability/risk studies across the globe have adopted static vulnerability assessment that represent drought vulnerability snapshot in time (Hagenlocher et al., 2019). Therefore, we used the static vulnerability indicators for the year 2010, 2060 and 2099 to quantify drought vulnerability for the baseline, Far future and Near future period respectively. In the case of drought risk assessment, the drought hazard capturing the droughts in baseline (1980-2015), Near (2021-2060) and Far (2061-2099) future period is combined with drought vulnerability at 2010, 2060 and 2099 respectively. This is clarified in the revised manuscript.

Line 336-337, Page 15

"The transition of drought vulnerability from one class of vulnerability from baseline to another class of vulnerability in the future is given in Table 3."

Line 375-376, Page 17

"The transition of drought risk from one class of vulnerability from baseline to another class of risk in the future is given in Table 4."

Line 129-134, Page 5

"In general, socio-economic development is a slow process, and takes time to reflect in terms of significant changes in the socio-economic indicators (Dellink et al., 2017). Further, majority of the drought vulnerability/risk studies across the globe have adopted static vulnerability assessment that represent drought vulnerability snapshot in time (Hagenlocher et al., 2019). Therefore, we used the static vulnerability indicators for the year 2010, 2060 and 2099 to quantify drought vulnerability for the baseline, Near future and Far future period respectively."

Line 231-232, Page 8

"Accordingly, the drought hazard capturing the droughts in baseline (1980-2015), Near (2021-2060) and Far (2061-2099) future period is combined with drought vulnerability at 2010, 2060 and 2099 respectively."

Figure 7 is mentioned only on two lines at the end of Section 3.3.1, could this summary be referred to in the previous discussions of hazard, vulnerability and risk as it is a more intuitive figure to understand than tables shown in Tables 2, 3 and 4.

Figure 7 is referred for the discussion on hazard, vulnerability and risk projections.

Line 397-401, Page 18

"Of all the future drought hazard scenarios considered, the RCP2.6-Far scenario revealed the largest area (2.8%) under high and very high hazard classes. In the case of drought vulnerability, as high as 42.9% area transits from lower vulnerability classes to higher vulnerability classes under RCP6.0-SSP2 Near future, with 93% area of the country under high and very high drought vulnerability class. Further, in the worst case drought risk scenario (RCP6.0-SSP2 Far future), it is observed that 2.7% area of the country is under high and very high drought risk class."

6) L233-238: Here you state that the RCP6.0-SSP2 Far future scenario is not the worst case for drought vulnerability, but was the most severe for drought risk due to the hazard component (L233-235). Then on L236-238 in the discussion of Figure 6, you state that drought vulnerability makes up the majority of the drought risk for the same scenarios. Please could you clarify these seemingly contradictory statements. I do not disagree with the point that we need more holistic drought risk assessment though.

Risk is an outcome of interaction between hazard and vulnerability, and is also a function of time. The fact that worst case scenarios are different for drought hazard and drought vulnerability, indicates dissimilar behavior of drought hazard and vulnerability indicators in inducing drought risk. For eg. population density is high in the Near future period (2060) as compared to the Far future (2100), while precipitation is continuously increasing in the projected period. A combination of such different hazard and vulnerability behavior in a given time period is effectively captured through comprehensive risk analysis. Therefore, though RCP6.0-SSP2 Far future scenario is not the worst-case scenario for drought vulnerability compared to RCP6.0-SSP2 Near future, interaction of high hazard with moderate to high vulnerability resulted in worst drought risk scenario in the case of RCP6.0-SSP2 Far future. However, in general, when the

change in drought risk for all the future scenarios are compared with the baseline, it is observed that area falling under drought risk due to drought vulnerability is increased (Figure 6). Therefore, though seemingly contradictory, the statements pointed out by the reviewer reflect realistic and to a large extent, expected drought risk behavior. Clarification on the above discussion is included in the revised manuscript.

Line 380-389, Page 17-18

"Risk is an outcome of interaction between hazard and vulnerability, and is also a function of time. The fact that worst case scenarios are different for drought hazard and drought vulnerability, indicates dissimilar behavior of drought hazard and vulnerability indicators in  inducing drought risk. For eg. population density is high in the Near future period (2060) as compared to the Far future (2100), while precipitation is continuously increasing in the projected period. A combination of such different hazard and vulnerability behavior in a given time period is effectively captured through comprehensive risk analysis. Therefore, though RCP6.0-SSP2 Far future scenario is not the worst-case scenario for drought vulnerability compared to RCP6.0-SSP2 Near future, interaction of high hazard with moderate to high vulnerability resulted in worst drought risk scenario in the case of RCP6.0-SSP2 Far future. However, in general, when the change in drought risk for all the future scenarios are compared with the baseline, it is observed that area falling under drought risk due to drought vulnerability is increased (Figure 6)."

7) Section 3.3.2: At what spatial scale is this information useful to policy makers? You could consider whether the gridded data used here is relevant to that of decision makers.

The usability of the developed hazard, vulnerability and risk maps in terms of their spatial scale, and potential applications by stakeholders and decision makers are elaborated in the revised manuscript.

Line 403-414, Page 18

"The drought hazard, vulnerability and risk projection maps from the present study, developed at $0.5°$ lat.$\times 0.5°$ lon spatial resolution are comparable with blocks/district level area. Therefore, these maps can assist the block-level administrators to know region-specific causative factors inducing

severe drought risk both in baseline and projected period, besides the indigenous components governing the drought risk. Also, these maps can inform the state or federal disaster management authorities concerning the climate action plans. The change in drought risk at different projected periods can modulate adaptation and mitigation strategies and can be included in decision support system for drought management. Since drought risk is found to be mainly driven by societal factors, action plans should be directed to improve socio-economic conditions. Groundwater conservation, conjunctive use of surface and groundwater, farmer participation in crop insurance, water saving farm practices and technologies are some important measures that can be adopted for raising the socio-economic standards. Further, the framework of our study is applicable for state-wise drought risk assessment with reliable hydro-climatic and socio-economic indicators. Such an assessment can recommend measures for watershed management, irrigation and agricultural practices and reorganizing water demand and supply management at a local scale."

8) L269-270: You say here that this study is an improvement for decision makers over existing drought risk assessment in India. Please briefly state what this improvement is.

A brief explanation of the improvement made in our study is now included in the revised manuscript.

Line 432-438, Page 21

"Drought risk projection studies undertaken over the Indian region are based on drought hazard alone, and no consideration has been given to the drought vulnerability component. The present study quantifies the relative contribution of drought hazard and drought vulnerability to the overall drought risk projections under a comprehensive risk framework. Thus, our analysis can aid different stakeholders involved in drought management for adaptation and mitigation plans under changing climate and socio-economic conditions. This marks the significant improvement of our study over existing studies on drought risk assessment in India under climate change."

**Supplementary information**

9) The description of the categorisation of the MSDI at the bottom of page 2 is not clear and should be expanded – for example, it should be clearly stated which categories from McKee et al., -

presumably extreme drought, severe drought, moderate drought etc., but you should include the thresholds and categories used in this study here. The meaning of the following two sentences ("Further, each category is organised into sub-groups based on the occurrence probabilities of the selected category. While the weightages are assigned to MSDI categories to account for drought magnitude, ratings are assigned to the sub-groups of each MSDI category to account for drought occurrence probability") are aren't sufficiently clear; the methodology of how weights were assigned should be described more clearly – for example, you start talking about ratings and clusters, and it is not clear what these are used for.

Our response to Reviewer #2, comment #2 addresses this comment.

10) The description of the drought vulnerability index is also overly complex and should be clarified by describing the process in words. You should also be specific for example on how exactly data were standardised and what is meant by 'suitable weights' – what makes them suitable, how have these be validated and checked.

Drought vulnerability assessment method is moved to main manuscript with detailed explanation starting from data standardization. Weights for categorical indicators are shown in Table S1. Further, our response to Reviewer #2, comment #2 clarifies the concerns from this comment.

**Minor issues**

11) In several places e.g. L119 and in the supplementary information you mention that data have been standardised, please explain how these data are standardised (e.g. across time or space).

Drought risk values are standardized spatially. This is clarified in the revised manuscript.

Line 237-240, Page 8

"Drought risk values computed using Eq 15 are further standardized spatially to obtain the Drought Risk Index (DRI). Standardization of drought risk at each grid is carried out using the equation

$$DRI = \frac{Rsik - Risk_{min}}{Risk_{max} - Risk_{min}} \qquad \dots (16)$$

Standardization is performed such that the values are distributed between 0 and 1, so as to classify different risk categories."

12) L99-100: '…and brought to a monthly time-scales.' This isn't clear – how and what metric? Total precipitation? Average soil moisture? Please expand on this comment.

The projected daily precipitation is cumulated over each month to get the monthly precipitation values and converted its units from kg m$^{-2}$ s$^{-1}$ to mm. The projected monthly soil moisture (average monthly soil moisture) from the model is converted from kg m$^{-2}$ to m$^3$/m$^3$. This is clarified in the revised manuscript.

Line 108-111, Page 4

"The projected daily precipitation is cumulated over each month to get the monthly precipitation values and converted its units from kg m$^{-2}$ s$^{-1}$ to mm. The projected monthly soil moisture (average monthly soil moisture) from the model is converted from kg m$^{-2}$ to m$^3$/m$^3$."

13) L139: 'the country-accumulated data of these hydro-climatic variables…' please clarify how these data were accumulated

Data is summed over all grids. This is clarified in the revised manuscript.

Line 263, Page 11

"The country-wide accumulated data (summed over all grids) of these hydro-climatic variables is shown in Figure 2."

14) L145: ' …many regions is less **severe** compared to the baseline period.' Word missing

This is corrected in the revised manuscript.

Line 273, Page 11

 "…many regions is less severe compared to the baseline period."

15) L168-169: It is not clear whether this choice of low skill GCMs was in the current study or Koutroulis et al and Cook et al. – note that this same comment is also made on L209-210.

We have now clarified these statements in the revised manuscript.

Line 303-305, Page 12

"Such contradicting observations are possibly due to selection of low-skill GCMs (Aadhar and Mishra, 2020) in Koutroulis et al. (2019) & Cook et al. (2020)."

Line 346-347, page 15

"Such contradicting observations in drought vulnerability is possibly due the choice of low-skill GCMs in Koutroulis et al. (2019)."

16) L193: Highest differences… → The biggest difference in…

Done.

Line 329, Page 15

"The biggest difference in land use land cover…"

17) L198: many region of the country **are** expected to

Done

Line 333-334, Page 15

"It is observed that many regions of the country are…"

18) L211: …sub-division-…

Done

Line 348, Page 15

"…meteorological sub-division scale…"

19) L211: Here you mention meteorological sub-divisions, what exactly does this mean, how were these defined?

Meteorological sub-divisions are the meteorologically homogenous regions identified by India Meteorological Department (Kelkar and Sreejith, 2020). This is clarified in the revised manuscript.

Line 348-349, Page 15

"Meteorological sub-divisions are the meteorologically homogenous regions identified by India Meteorological Department (Kelkar and Sreejith, 2020)."

20) L222: …expected to be under high risk to drought compared… → …expected to have a high drought risk compared…

Done.

Line 360, Page 17

"…are expected to have high drought risk compared…"

21) A mix of tenses seems to be used throughout, this should be reviewed, ensuing that the past tense is used where appropriate.

Noted and revised.

22) In some places 'Far Future' is capitalised and others it is not – you should ensure this is consistent.

Noted and revised.

**Tables and Figures**

23) Table 1: Please also included the last date that the factors were updated and the date range that was available. Some of the descriptions of the datasets are unclear, for example, Water Bodies Fraction and Groundwater – it is not clear what these data actually are, and what they're measuring. It would be useful to include the unit of each factor where appropriate, e.g. for 'Groundwater', 'Water Bodies Fraction'

Table 1 is revised as per the reviewer's suggestion, and included in the revised manuscript.

24) Table 2: I find these tables unintuitive and difficult to marry up with the description of results in the text. e.g. L149-15150: 'The future drought hazard assessment using the projected hydro-climatic variables revealed that more than 35% area of the country is expected to be under the low hazard class, as compared to 8% in the baseline period' – I can't seem to make any of the very low hazard boxes up to these numbers or the difference between them. You could consider adding a more detailed example walk through of what this table means. There is also no reference to or legend for the colour scheme used here.

We have referred to Figure 7 for these numbers as

Line 285-287, Page 11

 "The future drought hazard assessment using the projected hydro-climatic variables revealed that more than 35% area of the country is expected to be under the low hazard class, as compared to 8% in the baseline period (refer Table 2 and Figure 7)."

Caption for Table 2 is revised and explained as

"Table 2. Transition of drought hazard from baseline period to projected period. The value in each cell represents the change in % area of the country from one hazard class to another. Red color shows transition, and blue represents no transition."

A detailed explanation of the table is included.

Line 274-278, Page 11

"The hazard transition from the baseline to different scenarios is presented in Table 2. In the transition matrix we compute the % area of the country that transitioned from one hazard class to other, to quantify the effect of climate change. The upper triangle in the table represents % area transition from lower to higher hazard classes, the lower triangle represents % area transition from higher to lower hazard classes, and the diagonal elements represent % area with no transition."

25) Table 3 & 4: I assume these are all showing SSP2 scenarios, please clarify in the caption.

Tables are now updated in the revised manuscript indicating SSP2 scenarios.

Table 3. Transition of drought vulnerability from baseline period to projected period. The value in each cell represents the change in % area of the country from one vulnerability class to another. Red color shows transition, and blue represents no transition.

[Figure]

Table 4. Transition of drought risk from baseline period to projected period. The value in each cell represents the change in % area of the country from one risk class to another. Red color shows transition, and blue represents no transition.

[Figure]

26) Figure 1: it would be helpful to refer to this figure in the description of methods included in the Supplementary Information. The DHI and DVI should be directly referenced in this figure to clarify when these are calculated in the processing chain – including mention of the baseline period.

In the revised methodology have referred to Figure 1 in the calculation of DHI and DVI as

Line 180-181, Page 6

"The weightage and rating scheme is depicted graphically in Figure 1."

Line 220-221, Page 8

"Positive ($d_i^+$) and negative ($d_i^-$) separation measures for each region $i$ are computed based on $A^+$ and $A^-$ (also shown in Figure 1)."

27) Figure 4: There are some parts that are white, yet white is not mentioned in the legend.

White regions represent no data region.  Legend for Figure 4 is revised accordingly.

[Figure]

Figure 4. Multi-model ensemble drought vulnerability maps for the scenarios a) Baseline, b) RCP2.6-SSP2 Near future, c) RCP2.6-SSP2 Far future, d) RCP6.0-SSP2 Near future, e) RCP6.0-SSP2 Far future.

28) Figure 5: I recommend adding another colour to the legend here to highlight the higher risk areas (such as a dark red or similar). Do the classes used represent any specific categories of risk?

Color of the legend is changed as per the suggestion. The classes here represent 
[revised manuscript text omitted]

---

## Author Response (AR2)

**Responses and Actions taken on Reviewers' Comments**

**Journal**: Natural Hazards and Earth System Sciences

**Manuscript Reference No**.: nhess-2022- 18

**Title**: Evolution of multivariate drought hazard, vulnerability and risk in India under climate change.

**Authors**: Venkataswamy Sahana, Arpita Mondal

We thank the Reviewer for reviewing our revised manuscript and providing valuable feedback that have helped improve the quality of the work significantly. In this document, we provide a point by point response and actions taken on the comments and suggestions from the reviewers. (Figure, line, table and page numbers referred to in this document are with respect to the revised manuscript unless mentioned otherwise.)

**Responses to comments from Referee #1**

General comment

I think the authors presented a valuable and interesting study about the future development of the drought hazard, vulnerability and risk in India. The revision of the description of the datasets and the methods is satisfactory and the authors improved the manuscript significantly. Before the acceptance of the manuscript, I think the authors should consider the following minor comments and technical corrections. A revision of the English language is also recommended.

We thank the reviewer for the positive and constructive feedback on our work. We have implemented the minor and technical corrections suggested by the reviewer. Further, we have proof-read the manuscript again and corrected for grammar and language wherever necessary. We have addressed the comments provided by you in the below sections.

Minor comments

L refers to the lines in the revised manuscript with the highlighted changes.

1. L77: Please replace 'deficiencies' with 'data'.

Line 76, Page 3

"…analysis of precipitation as well as soil moisture data"

2. L84: Please replace 'frequencies' with 'resolution'.

Line 83, Page 3

"….to monthly resolution for the historical drought hazard assessment."

3. L102-107: The authors could discuss in Section 3 the limitations due to uncertainty in the simulated precipitation and, particularly, in soil moisture data. Similarly, the authors could discuss the uncertainty in the socio-economic scenarios.

We agree with the reviewer that the uncertainties in the simulated datasets may affect the drought risk assessment. We have now explained this point in the revised manuscript.

Lines 338-341, Page 13

"It is to be noted that the four GCMs considered in the present study for precipitation and soil moisture simulations are bias-corrected for precipitation, and covers more uncertainty in temperature and precipitation changes compared to other GCM subsets (McSweeney et al., 2015). However, inclusion of other skilled GCMs can account for wide range of uncertainty in the drought hazard assessment."

Lines 384-387, Pages 16-17

"Further, the socio-economic challenges for adaptation and mitigation in different SSP narratives are lead by different development pathways (O'Neill et al., 2017). Therefore, adoption of other SSPs in drought vulnerability assessment may unveil other plausible drought vulnerability projections."

4. L124: 'comprise sensitivity…' instead of 'comprises of'.

Line 119, Page 4

"…comprise sensitivity, exposure and adaptive capacity indicators."

5. L190: Unclear what the authors mean with 'deficiencies'. Please use another term such as 'deficit'.

Replaced.

Line 154, Page 5

"…based on the deficits in precipitation and soil moisture."

6. L197: 'are' instead of 'is'.

Line 198, Page 7

"…chosen grid are given in Table S2"

7. L199: Please explain the meaning of small 'r' and 's'.

Here 'r' and 's' represents the value of the random variables R and S respectively.

Lines 164-165, Page 6

"where $r$ and $s$ represents the value of the random variables R and S respectively, and $p$ represents the joint probability of the precipitation and soil moisture."

8. L201-204: Based on this sentence it is unclear how 'sk' and 'rk' were computed. Please explain.

$r_k$ and $s_k$ here denote the kth observation for precipitation and soil moisture respectively. The number of joint occurrences ($m_k$) of precipitation and soil moisture pair below $r_k$ and $s_k$ from the whole set of observations is used to calculate empirical joint probability for kth observation based on bivariate Gringorten plotting position (Gringorten, 1963). This is clarified in the revised manuscript.

Lines 166-169, Page 6

"$r_k$ and $s_k$ here denote the kth observation for precipitation and soil moisture respectively. The number of joint occurrences ($m_k$) of precipitation and soil moisture pair below $r_k$ and $s_k$ from the whole set of observations is used to calculate empirical joint probability for kth observation based on bivariate Gringorten plotting position (Gringorten, 1963)."

9. Tables 2 and 3: Please add in the caption the thresholds or the definition for the transition or the no transition. This should be explained in the main text as well.

The meaning of transition term adopted in Tables 2 is as follows. The baseline and projected scenarios of drought hazard are represented using five different classes – very low, low, medium, high and very high. Every region (grid) of the country may transit from one class in the baseline scenario to another class in the projected scenario, or remain in the same class for both baseline and projected scenario. This transition area is presented as percentage values in Figure 4. Similar definition of transition in vulnerability and risk classes are adopted and are shown in Figures 6 and 8 respectively. This is explained in the revised manuscript.

Lines 305-307, Page 12

"The baseline and projected scenarios of drought hazard are represented using five different classes – very low, low, medium, high and very high. Every region (grid) of the country may transit from one class in the baseline scenario to another class in the projected scenario, or remain in the same class for both baseline and projected scenario."

**References:**

Gringorten, I. I.: A plotting rule for extreme probability paper, J. Geophys. Res., 68(3), 813–814, doi:10.1029/JZ068i003p00813, 1963.

McSweeney, C. F., Jones, R. G., Lee, R. W., and Rowell, D. P.: Selecting CMIP5 GCMs for downscaling over multiple regions, Clim Dyn, 44, 3237–3260, https://doi.org/10.1007/s00382-014-2418-8, 2015.

O'Neill, B. C., Kriegler, E., Ebi, K. L., Kemp-Benedict, E., Riahi, K., Rothman, D. S., van Ruijven, B. J., van Vuuren, D. P., Birkmann, J., Kok, K., Levy, M., and Solecki, W.: The roads ahead: Narratives for shared socioeconomic pathways describing world futures in the 21st century, Global Environmental Change, 42, 169–180, https://doi.org/10.1016/j.gloenvcha.2015.01.004, 2017.